# *ChronoMagic-Bench*: A Benchmark for Metamorphic Evaluation of Text-to-Time-lapse Video Generation

**Shenghai Yuan[1], Jinfa Huang[3], Yongqi Xu[1], Yaoyang Liu[1], Shaofeng Zhang[5],**
**Yujun Shi[6], Ruijie Zhu[7], Xinhua Cheng[1,4], Jiebo Luo[3], Li Yuan[1,2,*]**

[1] Peking University, Shenzhen Graduate School, [2] Peng Cheng Laboratory,
[3] University of Rochester, [4] Rabbitpre Intelligence, [5] Shanghai Jiao Tong University,
[6] National University of Singapore, [7] University of California Santa Cruz

{yuanshenghai,chengxinhua,xuyongqi}@stu.pku.edu.cn, shi.yujun@u.nus.edu,
yaoyangliu319@gmail.com, sherrylone@sjtu.edu.cn, rzhu48@ucsc.edu,
yuanli-ece@pku.edu.cn, {jhuang90@ur,jluo@cs}.rochester.edu

## Abstract

We propose a novel text-to-video (T2V) generation benchmark, *ChronoMagic-Bench*[2], to evaluate the temporal and metamorphic knowledge skills in time-lapse video generation of the T2V models (e.g. Sora [9] and Lumiere [4]). Compared to existing benchmarks that focus on visual quality and text relevance of generated videos, *ChronoMagic-Bench* focuses on the models' ability to generate time-lapse videos with significant metamorphic amplitude and temporal coherence. The benchmark probes T2V models for their physics, biology, and chemistry capabilities, in a free-form text control. For these purposes, *ChronoMagic-Bench* introduces **1,649** prompts and real-world videos as references, categorized into four major types of time-lapse videos: biological, human creation, meteorological, and physical phenomena, which are further divided into 75 subcategories. This categorization ensures a comprehensive evaluation of the models' capacity to handle diverse and complex transformations. To accurately align human preference on the benchmark, we introduce two new automatic metrics, MTScore and CHScore, to evaluate the videos' metamorphic attributes and temporal coherence. MTScore measures the metamorphic amplitude, reflecting the degree of change over time, while CHScore assesses the temporal coherence, ensuring the generated videos maintain logical progression and continuity. Based on the *ChronoMagic-Bench*, we conduct comprehensive manual evaluations of eighteen representative T2V models, revealing their strengths and weaknesses across different categories of prompts, providing a thorough evaluation framework that addresses current gaps in video generation research. More encouragingly, we create a large-scale *ChronoMagic-Pro* dataset, containing **460k** high-quality pairs of 720p time-lapse videos and detailed captions. Each caption ensures high physical content and large metamorphic amplitude, which have a far-reaching impact on the video generation community. [3]

## 1 Introduction

Text-to-video (T2V) generative models [89, 88, 94, 96, 43, 23, 76, 69] have developed rapidly recently. As the number of models continues to grow, there is an urgent need for evaluation methods that align

---

[*]Corresponding author

[2]"Chrono" derives from the Greek word "chronos", which means "time".

[3]The source data and code are publicly available on https://pku-yuangroup.github.io/ChronoMagic-Bench.

38th Conference on Neural Information Processing Systems (NeurIPS 2024) Track on Datasets and Benchmarks.

Table 1: **Comparison of the characteristics of our ChronoMagic-Bench with existing T2V benchmarks.** Most of them only assess two dimensions: visual quality and text relevance.

| Benchmark | Type | Visual Quality | Text Relevance | Metamorphic Amplitude | Temporal Coherence |
|---|---|---|---|---|---|
| UCF-101 [68] | General | ✓ | ✓ | ✗ | ✗ |
| Make-a-Video-Eval [66] | General | ✓ | ✓ | ✗ | ✗ |
| MSR-VTT [83] | General | ✓ | ✓ | ✗ | ✗ |
| FETV [47] | General | ✓ | ✓ | ✗ | ✓ |
| VBench [28] | General | ✓ | ✓ | ✗ | ✓ |
| T2VScore [79] | General | ✓ | ✓ | ✗ | ✓ |
| ChronoMagic-Bench (Ours) | Time-lapse | ✓ | ✓ | ✓ | ✓ |

with human perception, accurately reflecting the specific strengths and weaknesses of each model, thereby enabling the community to more easily select architectures that meet their requirements.

However, the current T2V benchmarks [47, 68, 66, 83, 28, 30] primarily assess the capability of generating general videos instead of time-lapse videos, failing to reflect the extent of physical priors encoded by the models. Additionally, the evaluation metrics they use mainly focus on visual quality and textual relevance, from early metrics like FID [26], FVD [71], and CLIPScore [25] to more recent ones like UMTScore [47], T2VQA [33], and UMT-FVD [47], all of which overlook two other crucial aspects of videos: metamorphic amplitude and temporal coherence. These limitations hinder the development of T2V models in generating videos with rich physical content.

Due to the greater metamorphic amplitude and temporal coherence of time-lapse videos, they contain more physical priors compared to general videos [89]. Therefore, to address the aforementioned issues, we introduce a benchmark called *ChronoMagic-Bench* for Metamorphic Evaluation of Time-Lapse Text-to-Video Generation, which provides a comprehensive evaluation system for T2V. We specifically designed four major categories for time-lapse videos, including biological, human creation, meteorological, and physical, and extended these to 75 subcategories. Based on this, we constructed *ChronoMagic-Bench*, comprising 1,649 prompts and their corresponding reference time-lapse videos. As shown in Table 1, compared to existing benchmarks [66, 83, 47, 28, 79, 68], *ChronoMagic-Bench* emphasizes generating videos with high persistence and strong variation, i.e., metamorphic videos with high physical prior content. Additionally, we developed MTScore for evaluating metamorphic amplitude and CHScore for temporal coherence to address the deficiencies in evaluation metrics and perspectives. With *ChronoMagic-Bench*, we conducted comprehensive qualitative and quantitative evaluations of almost all open&closed-source T2V models, enabling analysis of their strengths and weaknesses. The results highlighted the weaknesses of these models, including (1) almost all models fail to generate time-lapse videos with large variations; (2) poor adherence to prompts, necessitating multiple inferences to achieve satisfactory results; (3) the visual quality of single frame may be high, but flickering may occur, indicating poor temporal coherence.

Furthermore, we have meticulously curated the dataset *ChronoMagic-Pro* to provide the community with the first large-scale T2V dataset specifically designed for time-lapse video generation with higher physical prior content. ChronoMagic-Pro stands out from previous T2V datasets [83, 2, 77, 16, 75] as it comprises time-lapse videos (e.g., ice melting and flowers blooming) characterized by strong physical characteristics, high persistence, and variability. Considering the domain differences between time-lapse videos and general videos, we proposed an automatic time-lapse video collection framework to maintain video purity and improve annotation quality.

The contributions of this work are as follows:

**i) New T2V Benchmark.** We introduce *ChronoMagic-Bench* for comprehensive evaluation of T2V models, focusing on visual quality, text relevance, metamorphic amplitude, and temporal coherence.

**ii) New Automatic Metrics.** We develop MTScore and CHScore, which align better with human judgment than existing metrics, for assessing metamorphic attributes and temporal coherence.

**iii) New Insights for T2V Model Selection.** Our evaluations using *ChronoMagic-Bench* provide crucial insights into the strengths and weaknesses of various T2V models.

**iv) Large-Scale Time-lapse Video-Text Dataset.** We create *ChronoMagic-Pro*, a dataset with **460k** high-quality 720p time-lapse videos and detailed captions, promoting advances in T2V research.

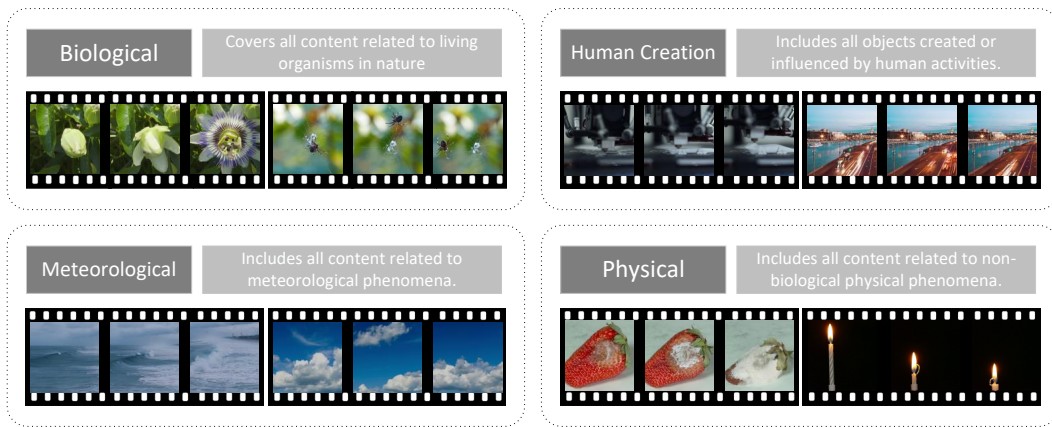

Figure 1: **Example of four major categories from ChronoMagic-Bench.** These categories fully encompass the physical world, allowing our benchmark and dataset to empower the community.

## 2 Related Work

**Automatic Metrics for Text-to-Video Generation.** Existing benchmarks [28, 34, 80, 37, 61] typically utilize Frechet Inception Distance (FID) [26], Frechet Video Distance (FVD) [71], CLIPScore [25], or their improved versions to assess the visual quality and text relevance of generated videos. For example, FETV [47] enhances FVD and CLIPScore within the UMT [39] feature space, resulting in UMT-FVD and UMTScore. Additionally, the CLIPScore feature extractor can be replaced with BLIP [92] to evaluate the relevance between text and generated content. To the best of our knowledge, existing T2V benchmarks [46, 47, 87, 18, 55] mainly assess these two aspects, with prompts based on general videos. This means that temporal coherence and metamorphic amplitude in videos have been overlooked, leading to the absence of automated metrics that indirectly reflect the physical content encoded by video models. Although [47, 28, 46] assess coherence, they are based on feature space or human evaluation, which is expensive and not sufficiently intuitive. Therefore, we propose the Metamorphic Score (MTScore) and Coherence Score (CHScore) to measure the metamorphic degree and temporal coherence of videos, filling this gap in the field.

**Datasets for Text-to-Video Generation.** Large-scale high-quality text-content pair data [10, 67, 56, 27] are essential for training generation models [93, 19, 59, 58, 50, 51, 52, 14, 54, 40, 6, 7, 24, 53, 91, 65]. To enable models to learn better representation spaces that simulate the real world, the larger the dataset and the richer the physical knowledge contained in the videos, the better the training effect. Researchers often construct these large-scale datasets through web scraping. For example, existing video generation models typically use WebVid-10M [2], which contains 10 million videos and captions. Recently released datasets, such as Panda-70M [16], HD-VG-130M [75], and InternVid [77], contain 70 million, 130 million, and 7.1 million text-video pairs, respectively. Despite their large sizes, these datasets consist of general videos with small metamorphic amplitude and short persistence of change, resulting in limited physical knowledge. Consequently, models trained on these datasets struggle to generate metamorphic videos. To address this issue, we propose the first large-scale dataset of time-lapse videos, comprising 460k 720P resolution video clips and their corresponding captions, which features strong persistence of changes, and high physical content.

## 3 ChronoMagic-Bench

### 3.1 Benchmark Construction

**Prompt Construction.** To comprehensively evaluate the time-lapse video generation capabilities of existing T2V models, the designed text prompts need to cover as many metamorphic types as possible, and the corresponding reference videos must be of relatively high quality. Manual construction is impractical; therefore, to build a T2V benchmark rich in visual concepts, we first manually created a search term database suitable for diverse and broadly applicable time-lapse videos. We then counted the number of videos obtainable for each search term and filtered them based on frequency, resulting in a search database containing 75 categories of time-lapse videos. Additionally, since there

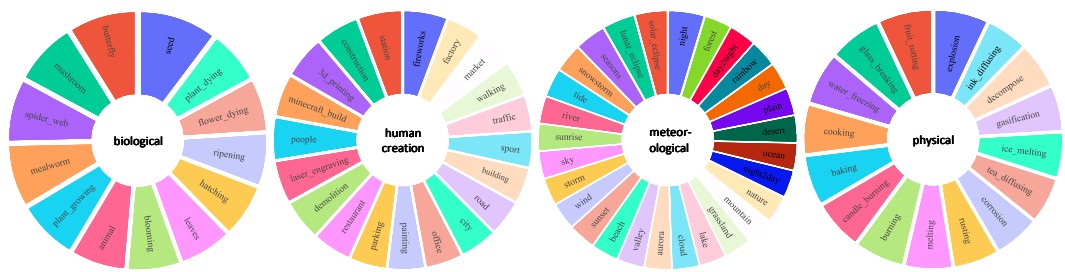

Figure 2: **Categories of Time-lapse Videos:** Firstly, we classify the videos into four major categories (biological, human creation, meteorological, physical), which are further subdivided into 75 subcategories (e.g. animal, parking, beach, melting).

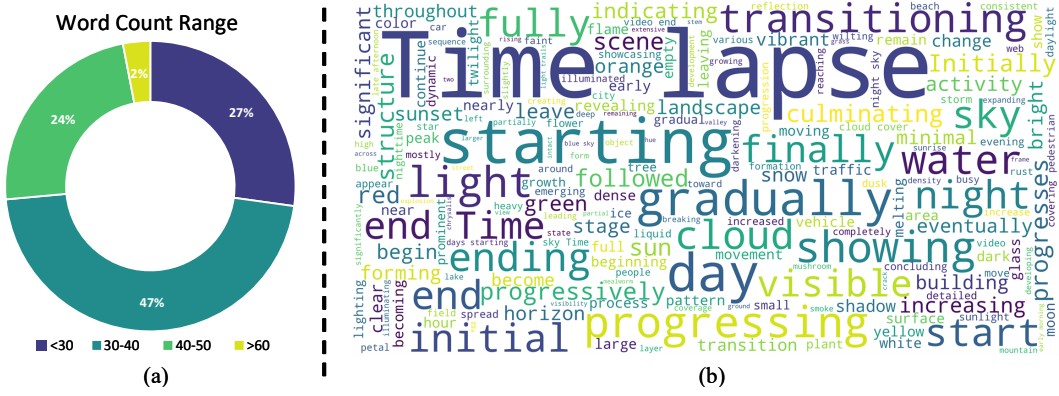

Figure 3: **The word cloud and word count range of the prompts in the ChronoMagic-Bench.** It shows that prompts mainly describe videos with large metamorphic amplitude and long persistence.

are four major nature phenomena: *biological* covers all content related to living organisms, such as plant growth, animal activities, microbial movement, etc. *Human creation* includes all objects created or influenced by human activities, such as the construction process of buildings, urban traffic flow, etc. *Meteorological* includes all content related to meteorological phenomena, such as cloud movement, storm formation, etc. *Physical* includes all content related to non-biological physical phenomena, such as water flow, volcanic eruptions, etc. We divide the 75 subcategories into four major categories (*biological*, *human creation*, *meteorological*, and *physical*), as shown in Figure 1. We then utilized a search engine to crawl 20 or more high-quality videos from video platforms for each category, ultimately gathering a total of 1,649 videos. Finally, we use GPT-4o [1] to accurately caption these videos and treat these captions as the text prompts for the benchmark. For more details about benchmark construction, please refer to Appendix E.

**Benchmark Statistics.**     We collect a total of 1,649 prompts with corresponding videos and categories, the specific data distribution is shown in Figure 2, indicating that 75 categories have a comparable number of test cases to reflect the time-lapse video generation capabilities of different models accurately. Each data sample in ChronoMagic-Bench consists of four elements: prompt $p$, reference video $v$, sub-category $c_1$, and major category $c_2$. Since existing T2V models typically use CLIP as the text encoder, which supports a maximum input of 77 tokens, we have limited the length of $p$ to within 77 tokens for general applicability, as shown in Figure 3(a). Although the length is limited, the diversity remains rich. By comparing the main words in the word cloud, as shown in Figure 3(b), it is observed that terms related to time-lapse videos such as "transitioning," "progressing," "increasing," and "gradually" appear most frequently. These terms significantly highlight ChronoMagic-Bench's focus on large metamorphic amplitude, strong persistence of changes, and high physical content. In addition, words from four major categories are distributed, such as biological (seed, butterfly, etc.), human creation (Minecraft, traffic, etc), meteorological (sunset, tide, etc), and physical (burning, explosion, etc). For detailed explanations of the 75 subcategories, please refer to the Appendix E.

---

**Algorithm 1** Calculation of Coherence Score

---

1: **Input:** Video, pre-trained model with grid size $G$ and threshold $T$
2: **Output:** Coherence score
3: Process input video using pre-trained model with grid size $G$ and threshold $T$ to get $p_{\text{vis}}$
4: **for** each frame $i$ **do**
5:     count the number of missing tracking points in each frame (except the time vanishing point)
6:     $m[i] \leftarrow \frac{1}{N} \sum_{j=1}^{N} (1 - p_{\text{vis}}[0, i, j])$
7: **end for**
8: **for** each frame $i$ **do**
9:     $\Delta m[i] \leftarrow |m[i+1] - m[i]|$
10:     **if** $\Delta m[i] > T$ **then**
11:         frame $i$ will be added to the set frames_to_be_cut
12:         $C_{\text{missed}} \leftarrow C_{\text{missed}} + \Delta m[i]$
13:     **end if**
14: **end for**
15: $R_{\text{cut}} \leftarrow \frac{\text{len(frames\_to\_be\_cut)}}{\text{frames}}$
16: $R_{\text{missed}} \leftarrow \frac{1}{\text{frames}} \sum_{i=1}^{\text{frames}} m[i]$
17: $V_{\text{missed}} \leftarrow \text{std}(\Delta m)$
18: $M_{\text{missed}} \leftarrow \max(\Delta m)$
19: $\text{C\_sum} \leftarrow \lambda_1 \hat{R}_{\text{missed}} + \lambda_2 \hat{V}_{\text{missed}} + \lambda_3 \hat{R}_{\text{cut}} + \lambda_4 \hat{C}_{\text{missed}} + \lambda_5 \hat{M}_{\text{missed}}$
20: $\text{Coherence\_score} \leftarrow \frac{1}{\text{C\_sum}}$

---

### 3.2 New Automatic Metrics

As previously mentioned, existing evaluation metrics mainly assess two aspects: visual quality and textual relevance, and the prompts only describe general videos. This indicates a lack of metrics for evaluating the capability to generate time-lapse videos, which not only need to measure the aforementioned two aspects but also need to assess metamorphic amplitude and temporal coherence.

**Metamorphic Score.** To the best of our knowledge, there is no existing automated evaluation metric for assessing metamorphic amplitude. A simple way is to use questionnaires or GPT-4o [1], which, although highly effective, is expensive. Another way is to use the open-source model [78], which, although less effective, is much cheaper. To address this, we propose both coarse-grained and fine-grained scores to measure the metamorphic amplitude, aiming to balance cost and performance.

For the coarse-grained score (i.e. MTScore), we initially designed $N$ retrieval sentences (please refer to Appendix B.1 for more details). We then input these sentences into a video retrieval model [78], resulting in the computation of probabilities for $n$ metamorphic and $m$ general videos. Let $P_i^{\text{meta}}$ and $P_i^{\text{gen}}$ represent the probabilities for the $i$-th metamorphic and general retrieval sentences, respectively. We then integrate these probabilities to derive a coarse-grained metamorphic score $S_c$:

$$S_c = \frac{\sum_{i=1}^{n} P_i^{\text{meta}}}{\sum_{i=1}^{n} P_i^{\text{meta}} + \sum_{i=1}^{m} P_i^{\text{gen}}} \tag{1}$$

Due to the strong instruction-following capability and world-understanding ability of GPT-4o, it can partially replace humans. For the fine-grained score (GPT4o-MTScore), we use GPT-4o as the evaluator. Specifically, we set a 5-point evaluation standard, then uniformly sample $T$ frames and input them into GPT-4o[1] to get the score. More implementation details are provided in Appendix B.

**Temporal Coherence Score.** Temporal coherence is crucial for time-lapse videos because they span a large time range. Current benchmarks assess coherence either through questionnaires [47] or by employing methods based on feature space calculations [28, 46]. The former approach is time-consuming, whereas the latter lacks intuitiveness and does not support visualization. Therefore, we developed the Coherence Score (CHScore) based on a video tracking model [29] as shown in Algorithm 1. More details are provided in the Appendix B.2. First, we process the input video using a pre-trained model with grid size $G$ and threshold $T$ to obtain $p_{\text{vis}}[i, j]$ (the visibility of point $j$ in frame $i$). Next, we count the number of missing tracking points $m[i]$ in each frame and the change in missed points between consecutive frames $\Delta m[i]$. To make the CHScore robust to the temporally coherent disappearance of points, we further calculate the direction of camera/object movement

Table 2: **Comparison of the statistics of our ChronoMagic-Pro with existing T2V datasets.**

| Dataset | # Categories | Video clips | Resolution | Type | Average length | Video duration (h) |
|---|---|---|---|---|---|---|
| MSR-VTT [83] | General | 10K | 240p | Video-Text | 15.0s | 40 |
| WebVid-10M [2] | General | 10M | 360p | Video-Text | 18.72s | 52K |
| InternVid [77] | General | 234M | 720p | Video-Text | 11.90s | 760.3K |
| Panda-70M [16] | General | 70M | 720p | Video-Text | 8.50s | 166.8K |
| HD-VG-130M [75] | General | 130M | 720p | Video-Text | 4.93s | 178K |
| Time-Lapse-D [81] | Time-lapse | 2K | 360p | Video | - | - |
| Sky Time-Lapse [85] | Time-lapse | 17K | 1080p | Video | - | - |
| ChronoMagic [89] | Time-lapse | 2K | 720p | Video-Text | 11.4s | 7 |
| ChronoMagic-Pro | Time-lapse | 460K | 720p | Video-Text | 234s | 30K |

based on the tracking points across all frames. If the tracking point j of frame i disappears in the far direction, it is not included in the calculation of $m[i]$. We then calculate the average proportion of missed points per frame $R_{\mathrm{missed}}$, indicating the overall visibility issue across the video. Following this, we compute the variation in the number of missed points between consecutive frames $V_{\mathrm{missed}}$, measuring frame-to-frame coherence. We also determine the ratio of frames that need to be cut $R_{\mathrm{cut}}$, reflecting the extent of video editing required, and count the number of consecutive changes in missed points exceeding the threshold $C_{\mathrm{missed}}$, indicating frequent large-scale instability in point tracking. Additionally, we measure the maximum continuous change in missed points $M_{\mathrm{missed}}$, highlighting the most severe continuity breaks in the video. Finally, we integrate these metrics to calculate the Coherence Score (CHScore). In the subsequent section, the actual CHScore is scaled by 0.1 to provide a more concise representation. Further details can be found in the appendix B.2.

### 3.3 Application Scope

ChronoMagic-Bench proposes automatic scores for measuring *metamorphic amplitude* and *temporal coherence*. When combined with existing metrics for *visual quality* and *textual relevance*, such as FVD [71], CLIPScore [25], UMT-FVD [47], and UMTScore [47], a comprehensive evaluation of T2V models across four dimensions can be achieved. Additionally, we can use human evaluation to more accurately assess these four dimensions.

## 4 ChronoMagic-Pro

**Multi-Aspect Data Curation.** As previously mentioned, existing large-scale text-video datasets primarily consist of general videos with limited physical information content, restricting open-source models [6, 66, 23] to generating only general videos rather than time-lapse videos. To address this, we construct the first large-scale time-lapse video dataset by collecting time-lapse videos based on the search terms outlined in Section 3.1, ultimately obtaining 66,226 original videos. Following the Panda70m method [16], we split these videos to produce 460K semantically consistent single-scene video clips. Finally, we utilize the video annotation strategy similar to MagicTime [89], replacing GPT-4V [1] with the open-source ShareGPT4Video [13] to reduce computational overhead while ensuring high-quality video captions. We conduct a verification experiment on the dataset in the Appendix D.3. More details about dataset construction are provided in Appendix C.

**Dataset Statistics.** We collected time-lapse videos from 75 categories manually set by the human, with proportions being roughly similar. Some samples can be found in the Appendix C.4. ChronoMagic-Pro is the first high-quality large-scale time-lapse T2V dataset, which contains more physical knowledge than general videos, as shown in Table 2. As shown in Figure 4, in terms of duration, more than half (53.3%) of the videos have a duration of 0-15 seconds, a quarter (27.1%) are longer than 60 seconds, 12.1% are between 15-30 seconds, and the remaining videos are distributed between 30-60 seconds. Regarding resolution, 97% are high resolution (720P), 2% are ultra-high resolution (1080P), and the remaining videos have lower resolutions ranging from 360P to 480P. As the number of words accepted by the text encoder increases, we require the generated captions to be as detailed as possible, with 95% of captions containing more than 100 words. For aesthetic score [64], 73% videos get high scores ranging from 4 to 6. 14% of the videos had aesthetic indicators exceeding 6, and only a small portion of the videos scored below 3. This indicates that the quality of most videos is high. For the word distribution of the generated captions, please refer to Appendix C.3. Similar to Figure 3, ChronoMagic-Pro mainly focuses on changes (gradually, progressing, increasing, etc.), processes spanning a large amount of time, such as flower blooming and ice melting.

Table 3: **Quantitative comparison with state-of-the-art T2V generation methods for the text-to-video task in ChronoMagic-Bench.** "↓" denotes lower is better. "↑" denotes higher is better. "†" denotes full parameters fine-tuning, "‡" denotes fine-tuning using the Magic Training Strategy [89].

| Method | Venue | Backbone | UMT-FVD↓ | UMTScore↑ | MTScore↑ | CHScore↑ | GPT4o-MTScore↑ |
|---|---|---|---|---|---|---|---|
| ModelScopeT2V [73] | Arxiv'23 | U-Net | 194.77 | 2.909 | 0.401 | 61.07 | 2.86 |
| ZeroScope [69] | CVPR'23 | U-Net | 227.02 | 2.350 | 0.400 | **99.67** | 2.09 |
| T2V-zero [30] | ICCV'23 | U-Net | 209.66 | 2.661 | 0.400 | 20.78 | 2.55 |
| LaVie [76] | Arxiv'23 | U-Net | **166.97** | 2.763 | 0.346 | 77.89 | 2.46 |
| AnimateDiff V3 [23] | ICLR'24 | U-Net | 197.89 | **2.944** | 0.467 | 70.85 | 2.62 |
| VideoCrafter2 [11] | Arxiv'24 | U-Net | 178.45 | 2.753 | 0.433 | 80.10 | 2.68 |
| MCM-MSLION [90] | Arxiv'24 | U-Net | 202.08 | 2.33 | 0.417 | 62.60 | 3.04 |
| MagicTime [89] | Arxiv'24 | U-Net | 257.56 | 1.916 | **0.478** | 81.82 | **3.13** |
| Latte [49] | Arxiv'24 | DiT | 192.12 | 2.111 | 0.363 | 68.68 | 2.20 |
| OpenSora 1.1 [96] | Github'24 | DiT | 195.43 | 2.678 | **0.444** | 73.98 | 2.52 |
| OpenSora 1.2 [96] | Github'24 | DiT | 166.92 | 2.781 | 0.375 | 51.60 | 2.56 |
| OpenSoraPlan v1.1 [43] | Github'24 | DiT | 188.53 | 2.421 | 0.327 | 68.52 | 2.19 |
| EasyAnimate V3 [82] | Arxiv'24 | DiT | 164.30 | 2.713 | 0.349 | **90.54** | 2.32 |
| CogVideoX-2B [86] | Arxiv'24 | DiT | **159.31** | **3.225** | 0.404 | 43.15 | **2.92** |
| OpenSoraPlan v1.1† | Ours | DiT | 185.72 | 2.753 | 0.341 | 49.85 | 3.03 |
| OpenSoraPlan v1.1‡ | Ours | DiT | **180.11** | **2.864** | **0.346** | **70.12** | **3.05** |

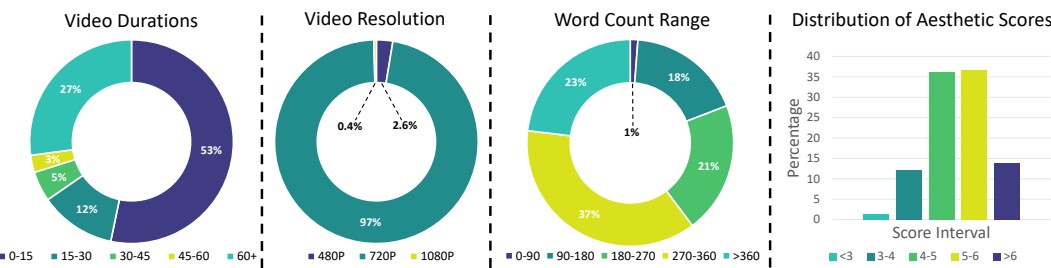

Figure 4: **Video clips statistics in ChronoMagic-Pro.** The dataset includes a diverse range of categories, clip durations and caption lengths, with most of the videos being in 720P resolution.

# 5 Experiments

## 5.1 Evaluation Models

We select fourteen open-source T2V models for evaluation, including both relatively advanced U-Net based models (e.g., ModelScopeT2V [73], ZeroScope [69], T2V-zero [30], LaVie [76], AnimateDiff [23], MagicTime [89], VideoCrafter2 [11] and MCM [90].) and emerging DiT-based models (e.g., Latte [49], OpenSoraPlan v1.1 [43], OpenSora 1.1 [96], OpenSora 1.2 [96], EasyAnimate [82], CogVideoX [86]). We also selected four closed-source models for evaluation, specifically **U-Net based**: *Gen-2* [63], *Pika-1.0* [36], **DiT-based**: *Dream Machine* [48], and *KeLing* [35]. All inference settings follow the official implementation. For more details, please refer to the Appendix D.

## 5.2 Evaluation Setups

**Evaluation Criteria.** We assess video quality primarily from the following four aspects: (a) **Visual Quality** measures the clarity, color saturation, contrast, and overall aesthetic effect, using UMT-FVD [47], an enhanced version of FVD [71]. (b) **Text Relevance** measures the correlation between the prompt and the video using UMTScore [47], an enhanced version of CLIPScore [25]. (c) **Metamorphic Amplitude** measures the diversity and dynamic changes in the video content, using the proposed Metamorphic Score. (d) **Temporal Coherence** measures the smoothness and logical sequence of the video content over time, using the proposed Coherence Score. Additionally, we use human evaluation to cross-verify the reliability of the four metrics.

**Implementation Details.** For each baseline, we generate corresponding triple results based on the 1,649 prompts contained in the ChronoMagic-Bench, resulting in 4,947 videos for each model. We then use the four automated metrics mentioned above to assess all the generated videos.

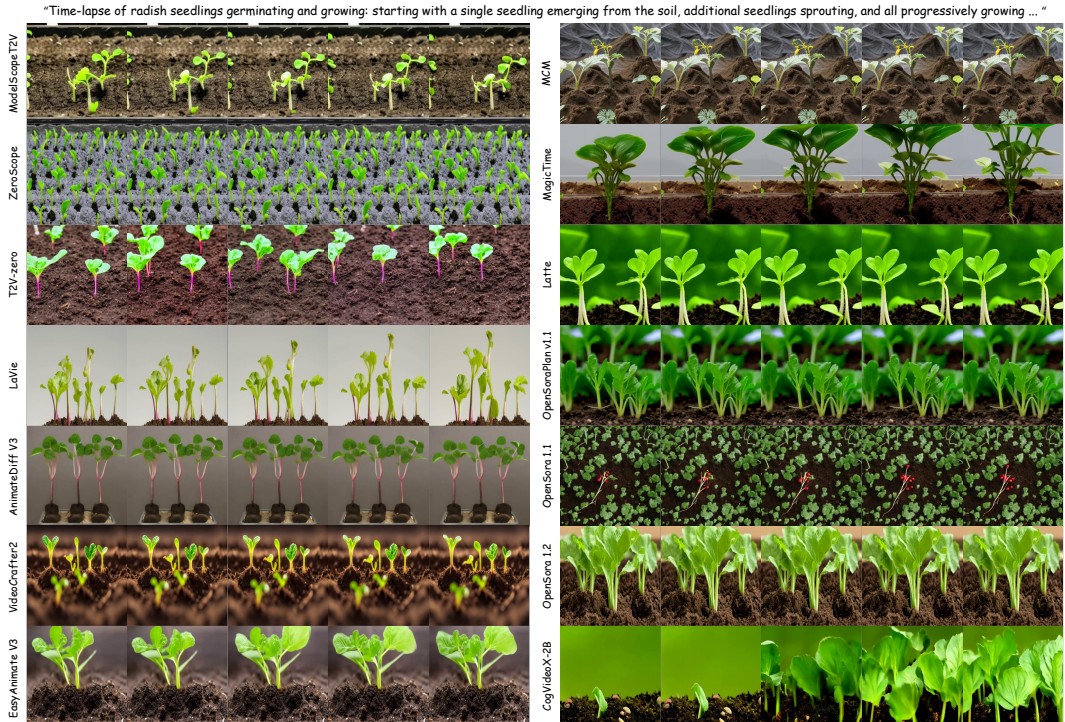

Figure 5: **Qualitative comparison with different T2V generation methods for the text-to-video task in ChronoMagic-Bench.** Most models can not follow instructions to generate time-lapse videos.

## 5.3 Comprehensive Analysis

**Quantitative Evaluation.** We first present and analyze the results from a qualitative perspective. All input texts are from our ChronoMagic-Bench. Unlike existing benchmarks that only assess general videos, our evaluation task focuses on generating metamorphic videos, such as the construction of houses in Minecraft, the blooming of flowers, the baking of bread rolls, and the melting of ice cubes. As shown in Figure 5, almost all U-Net-based and DiT-based models are limited to generating general videos and fail to follow prompts to produce videos with significant motion and temporal spans, except for MagicTime [89] and CogVideoX [86] (training data contains time-lapse videos), which underscores the importance of ChronoMagic-Pro dataset. Since T2V-Zero [30] is a zero-shot video generation model, its coherence is significantly lacking, although its visual quality is acceptable. Among the emerging DiT-based video models, CogVideoX [86] and OpenSora v1.2 [96] stands out as a representative that matches the performance of U-Net based methods, followed by EasyAnimate V3 [23], OpenSoraPlan v1.1 [43], while Latte [49] shows poor text-following capability.

**Qualitative Evaluation.** Next, we present and analyze the results of different T2V models from a qualitative perspective as shown in Table 3. Consistent with Figure 5, MagicTime [89] and CogVideoX [86], as the only model capable of generating metamorphic videos, has the highest MTScore and GPT4o-MTScore among all models. The other models, trained only on general videos, produce videos with limited motion range due to the minimal physical knowledge encoded in the models. It can also be seen that the results of the MTScore based on feature space with lower overhead and the GPT4o-MTScore based on question answering with higher overhead are roughly similar, proving the effectiveness of the proposed indicators. Additionally, ZeroScope [69] has limited metamorphic amplitude but the best coherence, while the zero-shot algorithm T2V-Zero [30] has the lowest CHScore. U-Net based and DiT-based models have similar CHScore, but the former shows superior average metamorphic amplitude. For visual quality and text relevance, the emerging CogVideoX [86] and EasyAnimate [82] performed best. The OpenSoraPlan v1.1 [43] and OpenSora 1.1&1.2 [96] have visual quality comparable to U-Net based methods, but slightly inferior text relevance. Only MagicTime [89] and CogVideoX [86] follows the prompt to generate

Table 4: **Quantitative Comparison with *Closed-Source* Generation Methods for the Text-to-Video Task in ChronoMagic-Bench-150.** To facilitate comparison under a unified standard, we also test *Open-Source* models. "↓" denotes lower is better. "↑" denotes higher is better.

| Method | Venue | Backbone | Status | UMT-FVD↓ | UMTScore↑ | MTScore↑ | CHScore↑ | GPT4o-MTScore↑ |
|---|---|---|---|---|---|---|---|---|
| Gen-2 [63] | Runway | U-Net | Close-Source | 218.99 | **2.400** | 0.373 | **125.25** | 2.62 |
| Pika-1.0 [36] | PikaLab | U-Net | Close-Source | 223.05 | 2.317 | 0.347 | 75.98 | 2.48 |
| Dream Machine [48] | LUMA | DiT | Close-Source | 214.91 | 2.387 | **0.474** | 95.97 | **3.11** |
| KeLing [35] | Kwai | DiT | Close-Source | **202.32** | 2.517 | 0.369 | 74.20 | 2.74 |
| ModelScopeT2V [73] | Arxiv'23 | U-Net | Open-Source | 230.74 | 2.783 | 0.409 | 61.01 | 3.01 |
| ZeroScope [69] | CVPR'23 | U-Net | Open-Source | 260.61 | 2.232 | 0.403 | **94.67** | 2.29 |
| T2V-zero [30] | ICCV'23 | U-Net | Open-Source | 250.22 | 2.559 | 0.399 | 18.54 | 2.62 |
| LaVie [76] | Arxiv'23 | U-Net | Open-Source | **210.39** | 2.714 | 0.350 | 81.32 | 2.50 |
| AnimateDiff V3 [23] | ICLR'24 | U-Net | Open-Source | 239.31 | **2.837** | 0.470 | 70.36 | 2.62 |
| VideoCrafter2 [11] | CVPR'23 | U-Net | Open-Source | 214.06 | 2.763 | 0.437 | 75.90 | 2.87 |
| MCM-MSLION [90] | Arxiv'24 | U-Net | Open-Source | 244.49 | 2.282 | 0.422 | 58.08 | **3.06** |
| MagicTime [89] | Arxiv'24 | U-Net | Open-Source | 294.72 | 1.763 | **0.479** | 77.98 | 3.05 |
| Latte [49] | Arxiv'24 | DiT | Open-Source | 232.29 | 2.122 | 0.366 | 72.57 | 2.42 |
| OpenSora 1.1 [96] | Github'24 | DiT | Open-Source | 241.09 | 2.676 | 0.448 | 75.94 | 2.57 |
| OpenSora 1.2 [96] | Github'24 | DiT | Open-Source | 210.93 | 2.681 | 0.383 | 51.87 | 2.50 |
| OpenSoraPlan v1.1 [43] | Github'24 | DiT | Open-Source | 228.70 | 2.459 | 0.331 | 61.50 | 2.21 |
| EasyAnimate V3 [82] | Arxiv'24 | DiT | Open-Source | 202.03 | 2.733 | 0.352 | **88.48** | 2.33 |
| CogVideoX-2B [86] | Arxiv'24 | DiT | Open-Source | **195.52** | **3.240** | **0.472** | 38.64 | **3.09** |

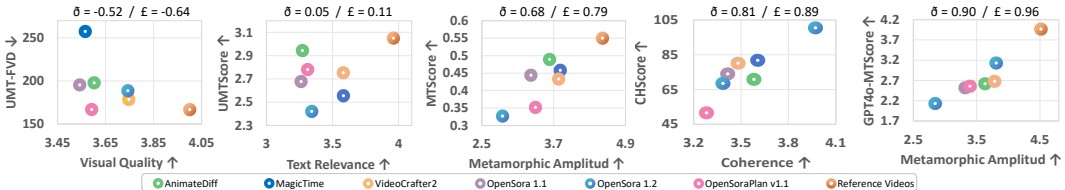

Figure 6: **Alignment between automatic metrics and human perception in terms of visual quality, textual relevance, metamorphic amplitude, and temporal coherence**. ð and £ represent Kendall↑ and Spearman↑ coefficients, respectively. ↑" denotes higher is better.

a time-lapse video, but the UMTScore [47] is the lowest. We infer that the UMT-FVD [47] and UMTScore [47] are inconsistent with human perception.

**Human Preference.** Finally, we cross-validate the effectiveness of the different metrics through Human Study. We randomly select the generated videos corresponding to 16 prompts and invited 212 participants to vote, obtaining manual evaluation results. To enhance user satisfaction, we select only six representative baseline results and reference videos from which users can choose. Figure 6 shows the correlation between automatic metrics and human perception. It is evident that the proposed three metrics, MTScore, CHScore, and GPT4o-MTScore, are consistent with human perception and can accurately reflect the metamorphic amplitude and temporal coherence of T2V models. Additionally, as mentioned earlier, UMTScore [47] cannot accurately measure text relevance, especially in the evaluation of time-lapse videos, where its Kendall and Spearman coefficients are the lowest. We infer that its feature space is not suitable for time-lapse video. For more details please refer to Appendix D.

**Extended Analysis of Closed-Source Models** In this section, we explore the performance and limitations of closed-source models. Given the impracticality of manually testing all 1,649 prompts in ChronoMagic-Bench, we selected two hard prompts from each of the 75 categories, resulting in ChronoMagic-Bench-150. We first analyze the results from a quantitative perspective. As shown in Table 4, with Dream Machine [48] performing better in metamorphic amplitude (MTScore, GPT4o-MTScore) and Pika-1.0 [36] showing the worst text relevance (UMTScore). DiT-based methods outperform U-Net based ones in visual quality. To facilitate comparison under a unified standard, we also test open-source models on ChronoMagic-Bench-150. It is evident that for most models, the MTScore and GPT4o-MTScore are low, and they are unable to generate videos involving complex state changes. Additionally, due to the inherent limitations of UMT-FVD [47] and UMTScore [47], they fail to accurately reflect the differences between open-source and closed-source models. However, the qualitative analysis across all models demonstrates that closed-source models consistently surpass open-source models in visual quality and textual relevance. Furthermore, it is worth noting that the results within the same domain (open/closed) align with human evaluations. We also conduct a detailed qualitative analysis, please refer to Appendix D.5 for more details.

**Exploratory Experiment on ChronoMagic-Pro.** To verify the validity and robustness of the ChronoMagic-Pro, we conducted quantitative validation based on OpenSoraPlan v1.1 [43]. Specifically, we fine-tuned the temporal module of the OpenSoraPlan v1.1 model using the Magic Training Strategy [89], based on the weights of OpenSoraPlan v1.1 [43]. Due to limited computational resources, we randomly selected only 10,000 video-text pairs from ChronoMagic-Pro for training. The results are shown in Table 3. After fine-tuning with ChronoMagic-Pro, the visual quality (i.e., UMT-FVD), text relevance (i.e., UMTScore), and metamorphic amplitude (i.e., MTScore and GPT4o-MTScore) were all effectively improved. Moreover, we utilized a straightforward method (e.g., full parameters) to fine-tune the model; however, the results suggest that this is less effective than the Magic Training Strategy [89]. More details and qualitative analysis are provided in Appendix D.3.

**Guideline for Model Selection.** With the increasing number of T2V models, the community faces challenges in selecting the most appropriate model due to the tendency of each model to showcase its best results. To address this issue, we provide a guideline for model selection based on the evaluation results of ChronoMagic-Bench: (a) Except for MagicTime [89], CogVideoX [86] and Dream Machine [48], most T2V models exhibit minimal metamorphic amplitude and cannot generate complete processes rich in physical changes, such as seed germination, sunrise, or building construction; (b) The visual quality of a single frame may be high, but when viewed in sequence, flickering often occurs, indicating poor temporal coherence. This issue is particularly evident in T2V-zero [30] and OpenSora 1.2 [96], whereas closed-source models do not exhibit this problem; (c) The emergence of Sora [9] has promoted the rapid development of DiT-based methods. Closed-source models based on DiT have comprehensively surpassed those based on U-Net. However, most open-source models' visual quality, text-following capability, and metamorphic amplitude still lag behind U-Net-based methods. We speculate that DiT-based models are more scalable and require more data, giving closed-source models a significant advantage over open-source models; (d) It is expensive to access massive data and computing resources. First, they can build datasets by crawling videos without copyright disputes. Second, adopting the U-DiT architecture may balance performance and cost to a certain extent; (e) Ordinary users who want to try T2V models can prioritize Dream Machine [48] and KeLing [35]. Researchers who wish to conduct in-depth research on T2V can prioritize the study of metamorphic video generation with CogVideoX [86], EasyAnimate [82], OpenSoraPlan [43] and OpenSora [96], as neither open-source nor closed-source models can achieve this function.

## 6  Conclusion

In this paper, we present ChronoMagic-Bench, the first benchmark specifically designed to assess the generation of time-lapse videos. It addresses the shortcomings of current benchmarks, which primarily focus on standard videos and overlook critical elements such as metamorphic amplitude and coherence. Additionally, we introduce two new automated metrics, MTScore and CHScore, which align with human perception. Based on ChronoMagic-Bench, we conduct a comprehensive evaluation of almost all open&closed-source leading text-to-video (T2V) models and provide crucial insights into the strengths and weaknesses of various models. Moreover, we propose ChronoMagic-Pro, the first large-scale time-lapse T2V dataset, to facilitate further research by the community.

## 7  Limitations and Future Work

While *ChronoMagic-Bench* offers a robust evaluation framework, there are still two limitations of this work. (1) The majority of the data is sourced from YouTube, where video quality varies significantly, necessitating extensive filtering based on aesthetic criteria, views, and likes. Additionally, most videos are licensed under CC BY 4.0, limiting their use to academic research. (2) Despite introducing MTScore and CHScore for metamorphic attributes and temporal coherence, a clear gap remains between these existing metrics and human preferences. Future efforts will focus on aligning automated metrics more closely with human judgment for a more accurate evaluation of T2V models.

## 8  Acknowledgments

We thank all the anonymous reviewers for their constructive comments. This work was supported in part by the Natural Science Foundation of China (No. 62202014, 62332002, 62425101), and Shenzhen Basic Research Program (No. JCYJ20220813151736001).

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

# NeurIPS Paper Appendix for *ChronoMagic-Bench: A Benchmark for Metamorphic Evaluation of Text-to-Time-lapse Video Generation*

## A    Related Works: Text-to-Video Generation Models

The emergence of large-scale text-to-image models [93, 60, 59, 58, 42, 5, 95, 14, 54, 40] has significantly advanced the field of Text-to-Video (T2V) generation [66, 6, 7, 21, 73, 91]. Existing T2V architectures can be categorized into two types: U-Net-based and DiT-based. The former typically builds on Stable Diffusion [62], extending the 2D U-Net to a 3D U-Net by adding temporal layers, thereby achieving high-quality video generation [74, 15, 23, 4, 11, 41]. The latter focuses on recreating open-source structures similar to Sora [9], using the DiT (Diffusion-Transformer) [57] framework for T2V generation [43, 96, 94, 20]. However, the generation quality of DiT-based architectures still lags behind that of U-Net-based architectures. MagicTime [89] notes that although these models have achieved basic video generation, the videos are typically limited to simple actions and scenes, resulting in the production of general videos rather than those enriched with physical priors like metamorphic/time-lapse videos. For a more intuitive representation, we have detailed a comparison of the metamorphic video generation capabilities of different algorithms.

# B   More details about Automatic Metrics

## B.1   Construction of retrieval sentences for Metamorphic Score

To obtain an effective Metamorphic Score (MTScore), we meticulously designed ten distinct retrieval texts to differentiate between time-lapse and normal videos. Although, in theory, only two retrieval sentences are needed to distinguish between general and time-lapse videos, multiple texts were used to enhance the model's robustness and accuracy. This approach also provides diverse linguistic representations for each video category, ensuring comprehensive coverage and minimizing bias. As shown in Table 5, the first five sentences (Index 0-4) describe general videos, capturing standard, unaltered video content in unique phrasings. The last five sentences (Index 5-9) describe time-lapse videos, characterized by accelerated playback or condensed time sequences, also phrased in various ways to capture different nuances. When calculating the MTScore, the video retrieval model uses these texts to evaluate each frame of the video, assigning probabilities based on the matches. The final result is obtained by summing the general probability and the metamorphic probability. For GPT4o-MTScore, we used a five-point rating scale and provided detailed scoring guidelines in the prompt, as shown in Table 6.

Table 5: **Retrieval sentences for coarse-grained score (MTScore)**

| Index | Sentence |
|---|---|
| 1 | A conventional video, not a time-condensed video. |
| 2 | A usual video, not an accelerated video sequence. |
| 3 | A normal video, not a time-lapse video. |
| 4 | A standard video, not a time-lapse. |
| 5 | An ordinary video, different from a fast-motion video. |
| 6 | A time-lapse video, distinct from a regular recording. |
| 7 | A time-lapse footage, not your typical video. |
| 8 | A fast-motion video, unlike a standard video. |
| 9 | A time-condensed video, not a conventional video. |
| 10 | An accelerated video sequence, not a usual video. |

Table 6: **Scoring Criteria for GPT4o-MTScore.** We set guidelines for each score to ensure that GPT-4o makes choices based on consistent criteria.

| Score | Brief Reasoning Statement |
|---|---|
| 1 | Minimal change. The scene appears almost like a still image, with static elements remaining motionless and only minor changes in lighting or subtle movements of elements. No significant activity is noticeable. |
| 2 | Slight change. There is a small amount of movement or change in the elements of the scene, such as a few people or vehicles moving and minor changes in light or shadows. The overall variation is still minimal, with changes mostly being quantitative. |
| 3 | Moderate change. Multiple elements in the scene undergo changes, but the overall pace is slow. This includes gradual changes in daylight, moving clouds, growing plants, or occasional vehicle and pedestrian movements. The scene begins to show a transition from quantitative to qualitative change. |
| 4 | Significant change. The elements in the scene show obvious dynamic changes with a higher speed and frequency of variation. This includes noticeable changes in city traffic, crowd activities, or significant weather transitions. The scene displays a mix of quantitative and qualitative changes. |
| 5 | Dramatic change. Elements in the scene undergo continuous and rapid significant changes, creating a very rich visual effect. This includes events like sunrise and sunset, construction of buildings, and seasonal changes, making the variation process vivid and impactful. The scene exhibits clear qualitative change. |

## B.2   Further Description of Temporal Coherence Score

We present a detailed description of the algorithm for computing the Temporal Coherence Score. Specifically, we first process input video using the pre-trained model with grid size $G$ and threshold $T$ to get visibility of point $p_{\text{vis}}$. Then, we count the number of missing tracking points $m[i]$ in each frame, and the change in missed points between consecutive frames $\Delta m[i]$:

$$m[i] \leftarrow \frac{1}{N} \sum_{j=1}^{N} (1 - p_{\text{vis}}[i, j]) \tag{2}$$

$$\Delta m[i] \leftarrow |m[i+1] - m[i]| \tag{3}$$

where $N = G \times G$, $i$ represents the position of the frame, $j$ identifies different tracking points, and $p_{\text{vis}}[i, j]$ indicates the visibility of point $j$ in frame $i$. To make the CHScore robust to temporally coherent disappearance of points, we first calculate the direction of camera/object movement based on the tracking points across all frames. Then, if the tracking point j of frame i disappears in the far direction, it is not included in the calculation of $m[i]$. Based on these, we then calculate the $R_{\text{missed}}$, which represents the average proportion of missed points per frame in the video. And the $V_{\text{missed}}$, which measures the variation in the number of missed points between consecutive frames, indicating frame-to-frame coherence:

$$R_{\text{missed}} = \frac{1}{F} \sum_{i=1}^{F} m[i] \tag{4}$$

$$V_{\text{missed}} = \sqrt{\frac{1}{F-1} \sum_{i=1}^{F-1} (\Delta m[i] - \bar{\Delta m})^2} \tag{5}$$

where $\Delta m[i] = m[i+1] - m[i]$, $\bar{\Delta m}$ is the mean of $\Delta m[i]$, $F$ is the total number of frames and $N$ is the number of points per frame. In addition, we need to calculate the $R_{\text{cut}}$, which indicates the ratio of frames that need to be cut to the total number of frames, reflecting the extent of video editing required. And the $C_{\text{missed}}$, which indicates the number of consecutive changes in missed points exceeding the threshold, indicating frequent large-scale instability in point tracking:

$$R_{\text{cut}} = \frac{|\{i : \Delta m[i] > T\}|}{F} \tag{6}$$

$$C_{\text{missed}} = \sum_{\substack{i=1 \\ \Delta m[i] > T}}^{F-1} \Delta m[i] \tag{7}$$

where $T$ is the threshold for significant missed point variation, and $|\{i : \Delta m[i] > T\}|$ represents the number of frames with significant missed point variation. Then we calculate the $M_{\text{missed}}$, which measures the maximum continuous change in missed points, reflecting the most severe continuity breaks in the video, and finally get the Coherence Score (CHScore):

$$M_{\text{missed}} = \max(\Delta m) \tag{8}$$

$$\text{CHScore} = \frac{1}{\lambda_1 \hat{R}_{\text{missed}} + \lambda_2 \hat{V}_{\text{missed}} + \lambda_3 \hat{R}_{\text{cut}} + \lambda_4 \hat{C}_{\text{missed}} + \lambda_5 \hat{M}_{\text{missed}}} \tag{9}$$

where $\hat{X}$ represents the normalized variable, and $\lambda_x$ denotes the corresponding weight coefficient. For the setting of $\lambda_1$ to $\lambda_5$, we follow the following principles. Specifically, $R_{\text{missed}}$ is a global metric representing the model's overall performance across the entire video and holds the highest significance, with a weight of $\lambda_1 = 0.35$. $V_{\text{missed}}$ measures the stability of missed points between frames, a critical aspect of video analysis, and is therefore assigned a weight of $\lambda_2 = 0.25$. $R_{\text{cut}}$ indicates abnormal situations and carries a weight of $\lambda_3 = 0.15$. $C_{\text{missed}}$, similar in function to the $R_{\text{cut}}$, serves as a secondary indicator, also weighted at $\lambda_4 = 0.15$. Lastly, $M_{\text{missed}}$ represents individual extreme cases and is assigned a lower weight of $\lambda_5 = 0.10$.

### B.3 Details of Temporally Coherent Disappearance of Points

The 'Temporally coherent disappearance of points' describes the phenomenon where tracking points vanish over time due to movements such as camera movement or water flow, potentially causing these points to exit the camera's field of view. To prevent this from influencing the CHScore calculation, we initially identify the direction of change for various points within the video, as depicted in Figure 7. Subsequently, points that vanish proximate to this directional endpoint are excluded from the CHScore calculation.

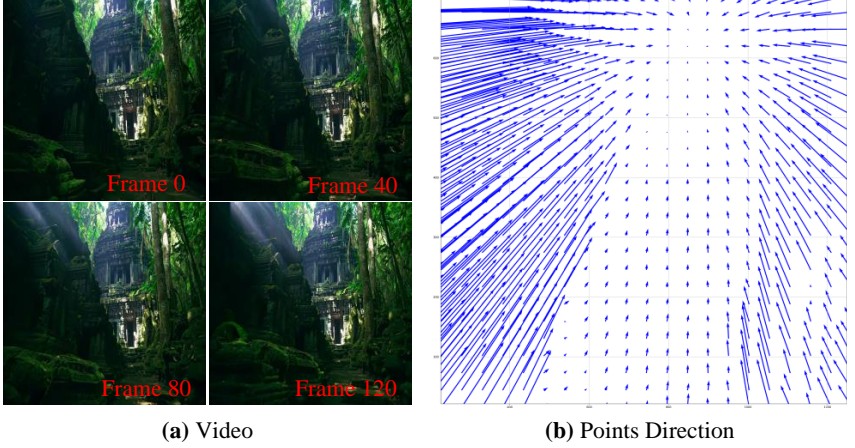

**(a)** Video                    **(b)** Points Direction

Figure 7: **The movement direction of the tracking points in the video.**

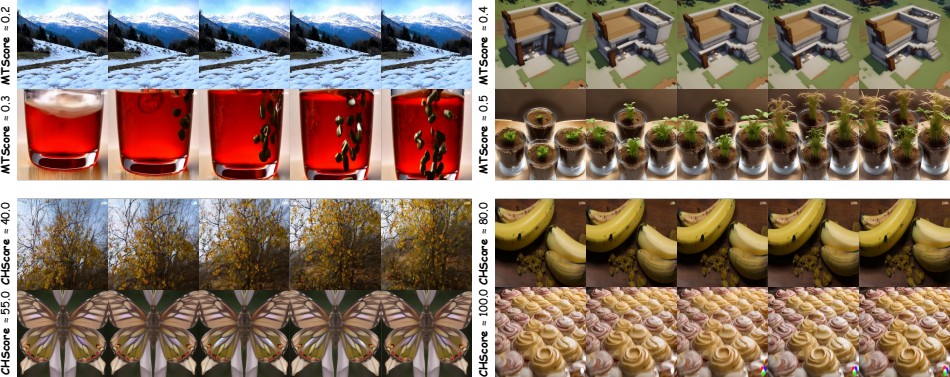

Figure 8: **Visual Reference for Varying Scores of MTScore and CHScore**. It is observed that higher scores correlate with increased metamorphic amplitude and coherence.

### B.4 Visual Reference of the Different Scores of MTScore and CHScore

We also provide some samples of different scoring magnitudes for MTScore and CHScore, as shown in Figure 8. It can be seen that both scores are consistent with human perception. We strongly recommend checking out the Project Page, which provides more case studies on the metrics.

## C More details about ChronoMaigc-Pro

### C.1 Data Preprocessing

Due to the abundance of low-quality videos on video platforms, we filter out lower-quality videos based on metadata such as view counts, comments, and likes after acquiring the original videos, ultimately obtaining 66,226 original videos. Additionally, since our training data is sourced from video platforms (e.g., YouTube) where videos are designed to engage the audience, they inherently contain many transitions (significant changes in content during video playback). To address this issue, we follow the method described in Panda70M [16] to split the videos into multiple semantically consistent single-scene clips. Specifically, OpenCV [8] initially splits the video by analyzing pixel differences between adjacent frames. Let $I_t$ be the image frame at time $t$; the difference between two adjacent frames can be computed as:

$$D_t = \sum_{i=1}^{H} \sum_{j=1}^{W} |I_t(i,j) - I_{t+1}(i,j)| \tag{10}$$

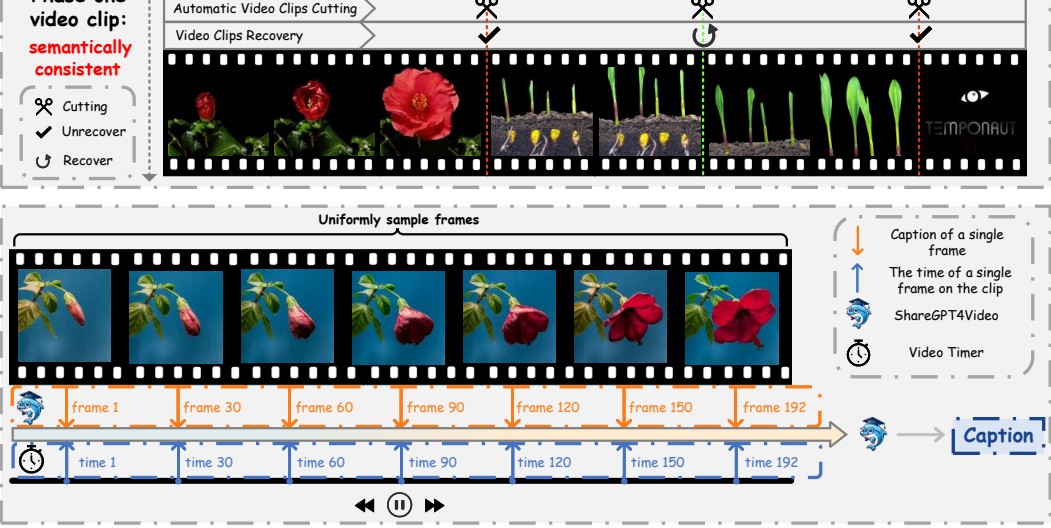

Figure 9: **The pipeline of constructing ChronoMagic-Pro.** *(Top)* We first use OpenCV [8] and ImageBind [22] to split the video and get semantically consistent single-scene video clips. *(Bottom)* Then, uniformly sample $N$ frames and obtain captions for each using ShareGPT4Video [13]. And finally let ShareGPT4Video [13] summarize the video caption based on these captions and their frame positions.

where $H$ and $W$ are the height and width of the frame, and $i$ and $j$ represent pixel positions, respectively. Videos are split into clips where $D_t$ exceeds a certain threshold $\tau$. Then, the ImageBind model [22] recombines erroneously split clips by analyzing feature space differences between adjacent clips. Let $\phi(I_t)$ represent the feature vector of frame $I_t$ obtained from the ImageBind model. The feature space difference between adjacent clips $C_i$ and $C_{i+1}$ can be computed as:

$$F_i = \left\| \phi(I_{t_i}) - \phi(I_{t_{i+1}}) \right\|_2 \tag{11}$$

where $t_i$ and $t_{i+1}$ are the times of the last frame of $C_i$ and the first frame of $C_{i+1}$, respectively. Clips are recombined where $F_i$ is below a certain threshold $\eta$. This process results in semantically consistent single-scene video clips.

### C.2 Time-Aware Annotation

After obtaining high-quality time-lapse video clips, it is crucial to add appropriate captions. The simplest approach is to input the video clips into a large multimodal model to generate text descriptions of the video content. However, our experiments found that the 8B [44], 13B [84], and 34B [38] models could not accurately describe the content of time-lapse videos, resulting in severe hallucinations, as shown in Figure 10. Therefore, we decided to follow the annotation strategy of MagicTime [89]. Unlike MagicTime, due to higher costs, we adopted an open-source model [13] instead of the closed-source GPT-4V [1]. As shown in Figure 10, we first uniformly sample $N$ frames from each video segment, input these $N$ frames into the multimodal large model to describe the content, and finally have the model summarize the final video captions based on the textual descriptions of $N$ frames and the corresponding position of each frame in the video. To balance cost and effectiveness, we chose to use the 8B multimodal large model [13] instead of the 34B.

### C.3 Distribution of the Generated Captions

To analyze the word distribution in our generated captions within ChronoMagic-Pro, we computed their frequency distributions. The results, shown in Figure 11, reveal a prevalence of terms related to time-lapse videos, including "change," "transition," and "progressing." Additionally, words from four primary categories are evident: biological (e.g., mealworm, flower, tree), human creation (e.g., building, painting, walking), meteorological (e.g., eclipse, cloud, sunrise), and physical (e.g., burning,

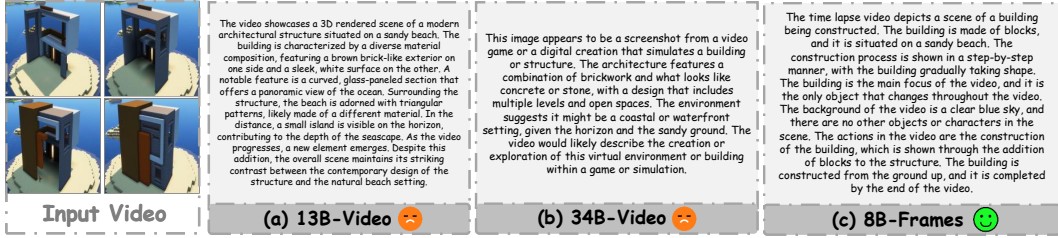

**Input Video**     (a) 13B-Video 😕     (b) 34B-Video 😠     (c) 8B-Frames 😊

Figure 10: **Ablation on different Captioning method.** Directly inputting the video into the model and having it describe the content is less effective than inputting keyframes into it.

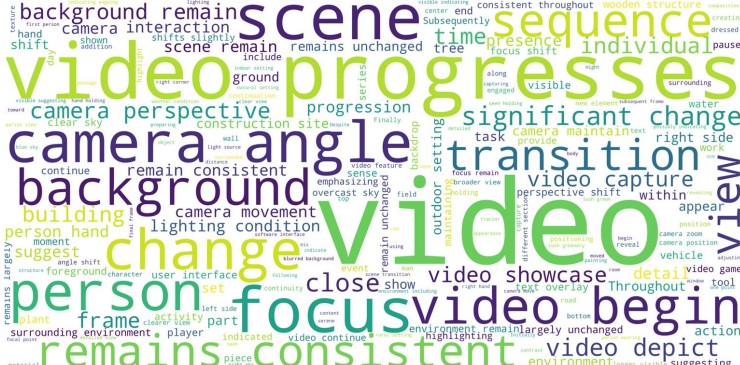

Figure 11: **The word clouds of the generated captions of ChronoMagic-Pro.** The dataset focuses on changes (gradually, progressing, increasing, etc.), processes spanning a large amount of time, such as flower blooming, ice melting, building construction, sunrise and sunset.

explosion). These terms underscore ChronoMagic-Pro's focus on large-scale metamorphic changes, persistent transformations, and substantial physical interactions.

### C.4 Samples of the ChronoMagic-Pro

Figure 12 showcases a diverse array of samples from the ChronoMagic-Pro dataset, which features an extensive collection of time-lapse videos across several categories, including plants, buildings, ice, food, and various other objects and phenomena. Each video captures dynamic changes over time, providing rich visual information that surpasses the physical knowledge contained in many existing Text-to-Video (T2V) datasets. These samples illustrate the dataset's diversity and depth, encompassing biological, human creation, meteorological, and physical categories, designed to support advanced research in high-dynamic text-to-video generation and related fields. Additionally, the dataset includes both time-lapse videos with significant state changes (e.g., flowers blooming) and videos with smaller state changes (e.g., clouds floating).

### C.5 Additional Statements

**1.** The aesthetic detector exhibits inherent biases, favoring artistic images, such as oil paintings and other art forms, over more realistic styles. Consequently, low aesthetic scores do not necessarily indicate poor-quality data. Retaining a small portion of such data can enhance the diversity of the videos. Thus, we include 13% of clips with low aesthetic scores in ChronoMagic-Pro.

**2.** There are two types of time-lapse videos: compressed and uncompressed. The former represents the entire process in a few seconds, while the latter can last for several minutes or even tens of minutes. ChronoMagic consists of compressed videos, whereas ChronoMagic-Pro includes both types to increase diversity. If the 60s+ videos, which account for 27% of ChronoMagic-Pro, are excluded, the average length would be only 12.36 seconds.

**3.** Since the data of ChronoMagic-Bench and ChronoMagic-Pro both include videos from YouTube, we deduplicate the data using video IDs. Additionally, we employed different annotation models

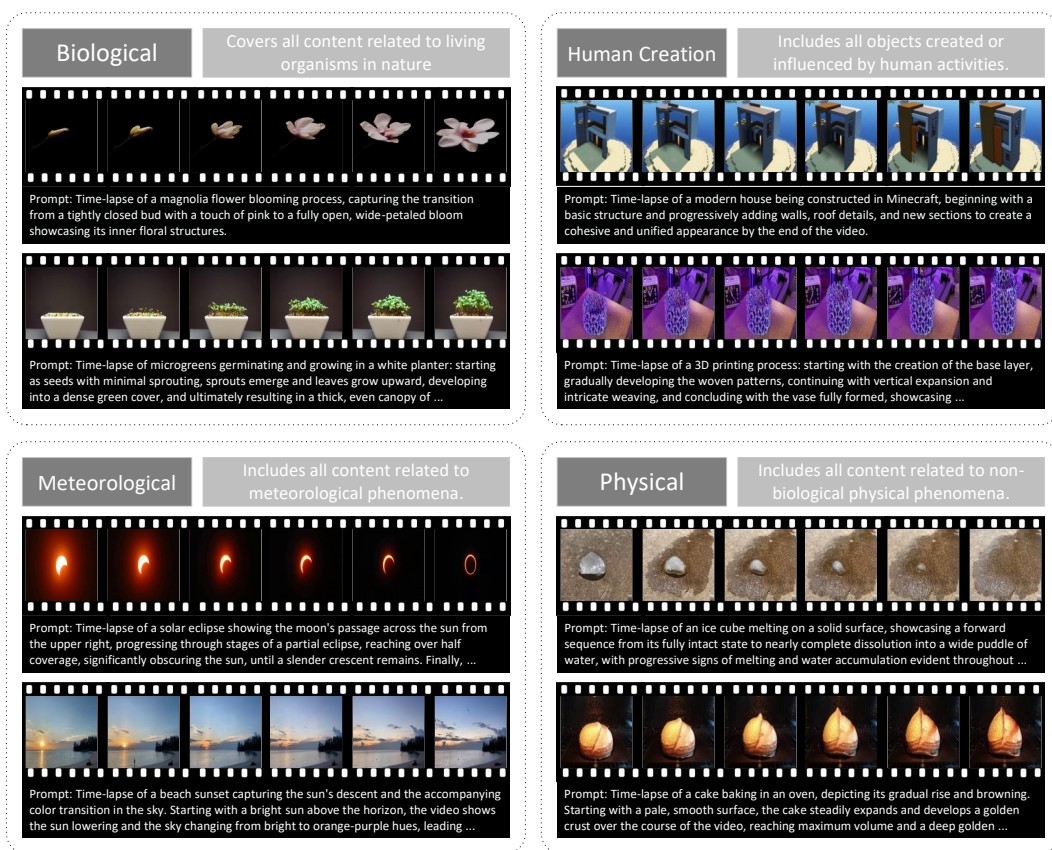

Figure 12: **Samples from the ChronoMagic-Pro dataset.** The dataset consists of time-lapse videos, which exhibit more physical knowledge than existing T2V dataset.

(e.g., GPT-4o [1], ShareGPT4Video [13]) to label the benchmark and dataset, further reducing the risk of data leakage.

# D  More Details about Experiment

## D.1  Details of Resource

We employ two types of GPUs: Nvidia H100 (x8) and Nvidia A800 (x8). All implementations are conducted based on the official code using the PyTorch framework.

## D.2  Details of Evaluation Models

Since most T2V models do not support dynamic resolution or variable duration, it is not feasible to standardize these parameters. Therefore, we follow the official popular settings [47, 28, 66, 80] to maintain a degree of fairness. Moreover, both MTScore and CHScore mitigate the influence of resolution by employing a resizing strategy that adjusts the shorter edge and utilizes center cropping. MTScore further employs a fixed frame extraction method to ensure a consistent frame count, while the different terms of CHScore are insensitive to *num_frames*, thereby mitigating discrepancies due to varying frame numbers.

**ModelScopeT2V.**  *Model Details.* ModelScopeT2V [73], featuring a U-Net architecture, extends the T2I model Stable Diffusion [62] by incorporating 1D temporal convolution and attention modules alongside the 2D modules for video modeling. Its training data consists primarily of image-text pairs (LAION [64]) and general video-text pairs (WebVid-10M [2] and MSR-VTT [83]), but it does not include the time-lapse videos discussed in this paper. *Implementation Setups.* We utilized the ModelScopeT2V code and model officially released on HuggingFace, maintaining the original

parameter settings. We used a spatial resolution of 256×256 and a frame rate of 8 fps to generate a 2-second (16-frame) video.

**ZeroScope.** *Model Details.* ZeroScope [69] is a watermark-free U-Net-based video model built on ModelScopeT2V [73], capable of generating high-quality 16:9 compositions and smooth video outputs. The model is trained on 9,923 clips and 29,769 labeled frames (24 frames per clip, 576×320 resolution) derived from the original weights of ModelScopeT2V [73]. The official documentation does not specify the exact training data; we speculate that time-lapse videos were not included. *Implementation Setups.* We utilized the ZeroScope_v2_576w code and model officially released on HuggingFace, maintaining the original parameter settings. We used a spatial resolution of 576×320 and a frame rate of 8 fps to generate a 3-second (24-frame) video.

**T2V-Zero.** *Model Details.* Text2Video-Zero [30], featuring a U-Net architecture, is a zero-shot video generation method based on the T2I model Stable Diffusion [62]. It generates latent codes for all frames using rich motion dynamics and utilizes a self-attention mechanism to enable all frames to interact with the latent codes of the first frame. This process ultimately achieves high spatial and temporal consistency in the video through denoising. It does not require training data and, therefore, does not use time-lapse videos as training data. *Implementation Setups.* We utilized the officially released Text2Video-Zero code and model, maintaining the original parameter settings. Specifically, we used the dreamlike-photoreal-2.0 version of Stable Diffusion [62], with a spatial resolution of 512×512 and a frame rate of 8 fps, to generate a 2-second (16-frame) video.

**LaVie.** *Model Details. Model Details.* LaVie [76], featuring a U-Net architecture, is an extension of the T2I model Stable Diffusion [62]. It converts the T2I model into a T2V model by adding temporal dimension attention after the spatial modules and adopting an image-video joint training strategy. Its training data primarily consists of image-text pairs (LAION [64]) and general video-text pairs (WebVid-10M [2] and Vimeo25M [76]), but it does not include the time-lapse videos discussed in this paper. *Implementation Setups.* We used the officially released LaVie code and model. Although LaVie [76] provides options for frame interpolation and super-resolution after video generation, we did not use them to maintain fairness. We followed the original parameter settings, using a spatial resolution of 512×320 and a frame rate of 8 fps, to generate a 2-second (16-frame) video.

**AnimateDiff.** *Model Details.* AnimateDiff [23], featuring a U-Net architecture, is an extension of the T2I model Stable Diffusion [62]. It attaches a newly initialized motion modeling module to a frozen text-to-image model, then trains it on video clips to extract reasonable motion priors for video generation. Its training data primarily consists of general video-text pairs (WebVid-10M [2]), excluding the time-lapse videos discussed in this paper. *Implementation Setups.* We used the officially released AnimateDiffV3 code and model, maintaining the original parameter settings. We used a spatial resolution of 384×256 and a frame rate of 8 fps to generate a 2-second (16-frame) video.

**MCM.** *Model Details.* MCM [90], featuring a U-Net architecture, is a distillation video generation method based on the T2I model Stable Diffusion [62]. It propose motion consistency models (MCM) to improve video diffusion distillation by disentangling motion and appearance learning, addressing frame quality issues and training-inference discrepancies. Its training data primarily includes image-text pairs (LAION-aes [64]) and general video-text pairs (WebVid-2M [2]), but it does not include the time-lapse videos discussed in this paper. *Implementation Setups.* We used the officially released MCM-modelscopet2v-laion code and model, maintaining the original parameter settings. We used a spatial resolution of 256×256 and a frame rate of 7 fps to generate a 2-second (14-frame) video.

**MagicTime.** *Model Details.* MagicTime [89] is a U-Net-based metamorphic video generation model built on AnimateDiff [23]. It is capable of generating time-lapse videos with significant time spans and pronounced state changes, such as the entire process of a seed blooming or building construction. The model is trained using 2,265 metamorphic (time-lapse) clips and the original weights from AnimateDiffV3 [23]. Its training data primarily includes ChronoMagic [89], making it the only existing T2V model that uses time-lapse videos in the training process. *Implementation Setups.* We used the officially released MagicTime code and model, maintaining the original parameter settings. We used a spatial resolution of 512×512 and a frame rate of 8 fps to generate a 2-second (16-frame) video.

**VideoCrafter2.** *Model Details.* VideoCrafter2 [11], featuring a U-Net architecture, is similar to AnimateDiff [23], as both add temporal modules to Stable Diffusion [62] to achieve video generation. However, VideoCrafter2 differs by encoding fps as a condition into the model and implementing

the I2V function. Its training data primarily includes image-text pairs (LAION-COCO [17], JDB [70]) and general video-text pairs (WebVid-10M [2]), but it does not include the time-lapse videos discussed in this paper. *Implementation Setups.* We used the officially released VideoCrafter2 code and model, maintaining the original parameter settings. We used a spatial resolution of 512×320 and a frame rate of 10 fps to generate a 2-second (20-frame) video.

**Latte.** *Model Details.* Latte [49] is a pioneer in open-source DiT-based T2V algorithms. It inherits the pure Transformer architecture of the T2I algorithm PixArt-$\alpha$ [12] and extends it by adding temporal modules after each spatial module, training from the original weights of PixArt-$\alpha$ [12] to achieve a DiT-based T2V algorithm. Its training data primarily includes general video-text pairs (Vimeo25M [76] and WebVid-10M [2]). Although it includes the time-lapse videos mentioned in this paper, they primarily consist of sky videos with fewer physical priors, making it unable to generate videos such as seed germination and flower blooming. *Implementation Setup.* We used the officially released LatteT2V code and model, maintaining the original parameter settings. We used a spatial resolution of 512×512 and a frame rate of 8 fps to generate a 2-second (16-frame) video.

**OpenSoraPlan v1.1.** *Model Details.* OpenSoraPlan v1.1 [43] is a high-quality video generation model based on Latte [49]. It replaces the Image VAE [31] with Video VAE (CausalVideoVAE [43]), similar to Sora [9], enabling the generation of videos up to approximately 21 seconds long and high-quality images. Its training data consists of videos and images scraped from open-source websites under the CC0 license, labeled using ShareGPT4Video [13] to create a high-quality self-built dataset. The official documentation does not specify the exact training data; we speculate that time-lapse videos were not used. *Implementation Setup.* We used the officially released OpenSoraPlan v1.1 code and model. Although it provides T2V models in three versions: 65 frames, 221 frames, and 513 frames, we chose the 65-frame version to ensure fairness by maintaining a similar video length to other models. We kept the original parameter settings, using a spatial resolution of 512×512 and a frame rate of 24 fps to generate a 3-second (65-frame) video.

**OpenSora 1.1 & 1.2.** *Model Details.* OpenSora 1.1 & 1.2 [96] is a high-quality DiT-based T2V model that introduces the ST-DiT-2 architecture, building on Latte [49]the former is based on the Diffusion Model and the latter is based on the Flow Model. It supports the generation of images or videos with any aspect ratio, different resolutions, and durations. Its training data consists of images and videos scraped from open-source websites and a labeled self-built dataset. The official documentation does not specify the exact training data; we speculate that time-lapse videos were not used. *Implementation Setup.* We used the officially released OpenSora 1.1 & 1.2 code and model. For OpenSora 1.1, we employed the stage-3 checkpoint, setting the spatial resolution to 512×512 and the frame rate to 24 fps, to generate a 2-second (48-frame) video. For OpenSora 1.2, we set the spatial resolution to 1280×720 and the frame rate to 24 fps, producing a 4-second (96-frame) video.

**CogVideoX** *Model Details.* CogVideoX [86] is a state-of-the-art text-to-video diffusion model that builds upon the success of large-scale DiT models. To enhance text-video alignment, CogVideoX utilizes an expert transformer with expert adaptive LayerNorm, facilitating deep fusion between modalities. The model implements 3D full attention to comprehensively model videos along both temporal and spatial dimensions, ensuring temporal consistency and capturing large-scale motions. Its training data consists of scraped videos and images, and custom refined Panda70M [16], COCO caption [45] and WebVid [2]. *Implementation Setup.* We use the officially released CogVideoX code and model. For our experiments, we set the spatial resolution to 720x480, generated 48 frames, and used a frame rate of 8 fps, resulting in a 6-second video.

**EasyAnimate** *Model Details.* EasyAnimate [82] is an advanced text-to-video generation model designed to create high-quality animated videos from textual prompts. It adopts U-ViT [3] architectures and slice-vae to avoid unstable training. Its training data consists of videos and images scraped from open-source websites, and open-source dataset 10M SAM [32] and 2M JourneyDB [70]. The official documentation does not specify that time-lapse videos were used. *Implementation Setup.* We utilized the officially released EasyAnimateV3 code and model. For our experiments, we used the 720P version of the model. As per the default setting, we set the spatial resolution to 1008x576, generated 96 frames, and used a frame rate of 24 fps, resulting in a 4-second video.

### D.3  Further Verification Experiment on ChronoMagic-Pro

Notably, after fine-tuning in ChronoMagic-Pro, the enhancement in metamorphic amplitude endowed OpenSoraPlan [43] with the ability to generate time-lapse videos of significant state changes, such as

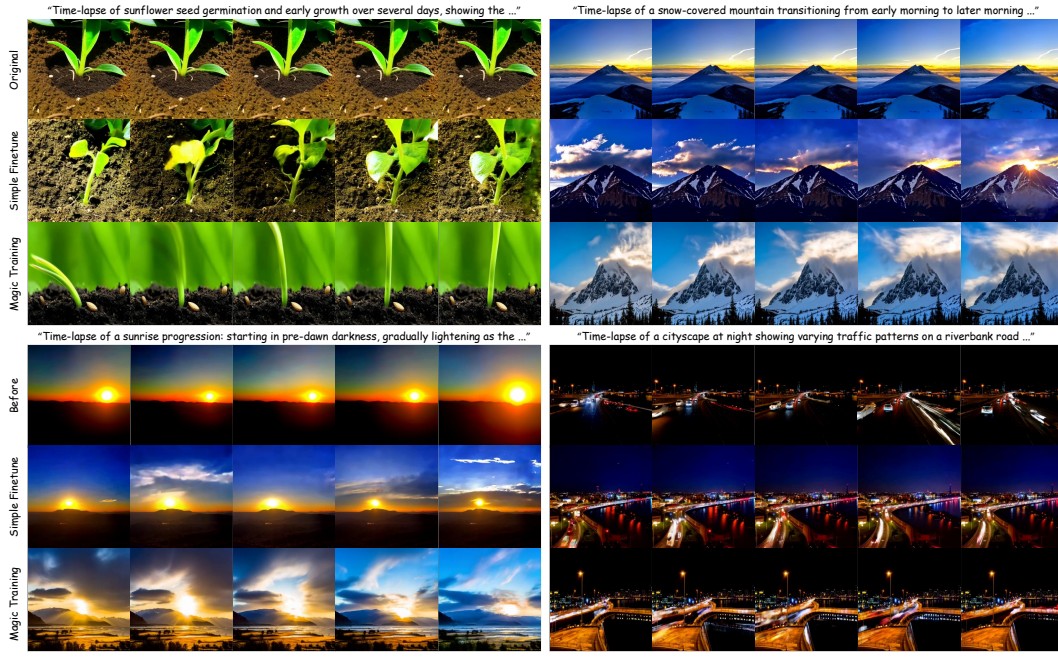

Figure 13: **Qualitative comparison of OpenSoraPlan v1.1 [43] before and after fine-tuning using ChronoMagic-Pro 10K.** After fine-tuning, the changes in the generated videos are no longer limited to lighting and camera movement, but are extended to changes in the state of objects. Additionally, it ensures that the *visual quality*, *text relevance*, and *coherence* are maintained without loss. Moreover, the efficacy of simple fine-tuning is inferior to that achieved through the Magic Training Strategy[89].

blooming flowers and city traffic. We provide additional qualitative analysis, as shown in Figure 13. It is evident that, after fine-tuning, the generated videos can extend changes beyond mere lighting and camera movements to alterations in the state of objects, while ensuring that the visual quality, text relevance, and coherence remain uncompromised. This proves that ChronoMagic-Pro can support existing models in generating high-quality time-lapse videos with significant state changes, providing a new approach for future T2V model training. Moreover, our findings suggest that with appropriate fine-tuning, it is possible to correct the common tendency of video models to produce nearly static videos on arbitrary topics. This phenomenon has also been observed in MagicTime-DiT [89], despite utilizing only around 2,000 time-lapse videos. However, it is important to note that the Magic Training Strategy [89], originally designed for U-Net-based models, may not be as effective for DiT-based models. In this study, we employ this methods solely for verification experiments. Additionally, the efficacy of simple fine-tuning is inferior to that achieved through the Magic Training Strategy [89].

### D.4 More Qualitative Evaluation on ChronoMagic-Bench

Due to space limitations, additional time-lapse videos generated by different baseline methods are shown in Figure 14. Similar to the results in the main text, most algorithms, except for MagicTime [89], fail to generate time-lapse videos with significant state changes, such as building construction. However, for time-lapse videos with smaller state changes, essentially faster-moving videos like city traffic changes, U-Net-based methods [73, 69, 30, 76, 23, 11, 89] exhibit much better visual quality, text relevance, and coherence compared to DiT-based methods [49, 43, 96]. This again demonstrates that U-Net-based methods are currently more stable and capable of producing satisfactory results with minimal inference. All videos generated by all models on ChronoMagic-Bench is publicly available on https://pku-yuangroup.github.io/ChronoMagic-Bench.

### D.5 More Analysis of Closed-Source Models

We present and analyze the results from a qualitative perspective, as shown in Figure 15. The results are consistent with Table 4. For metamorphic amplitude, most methods can only generate simple time-lapse videos, such as traffic flow; only Dream Machine [48] can generate a moderately

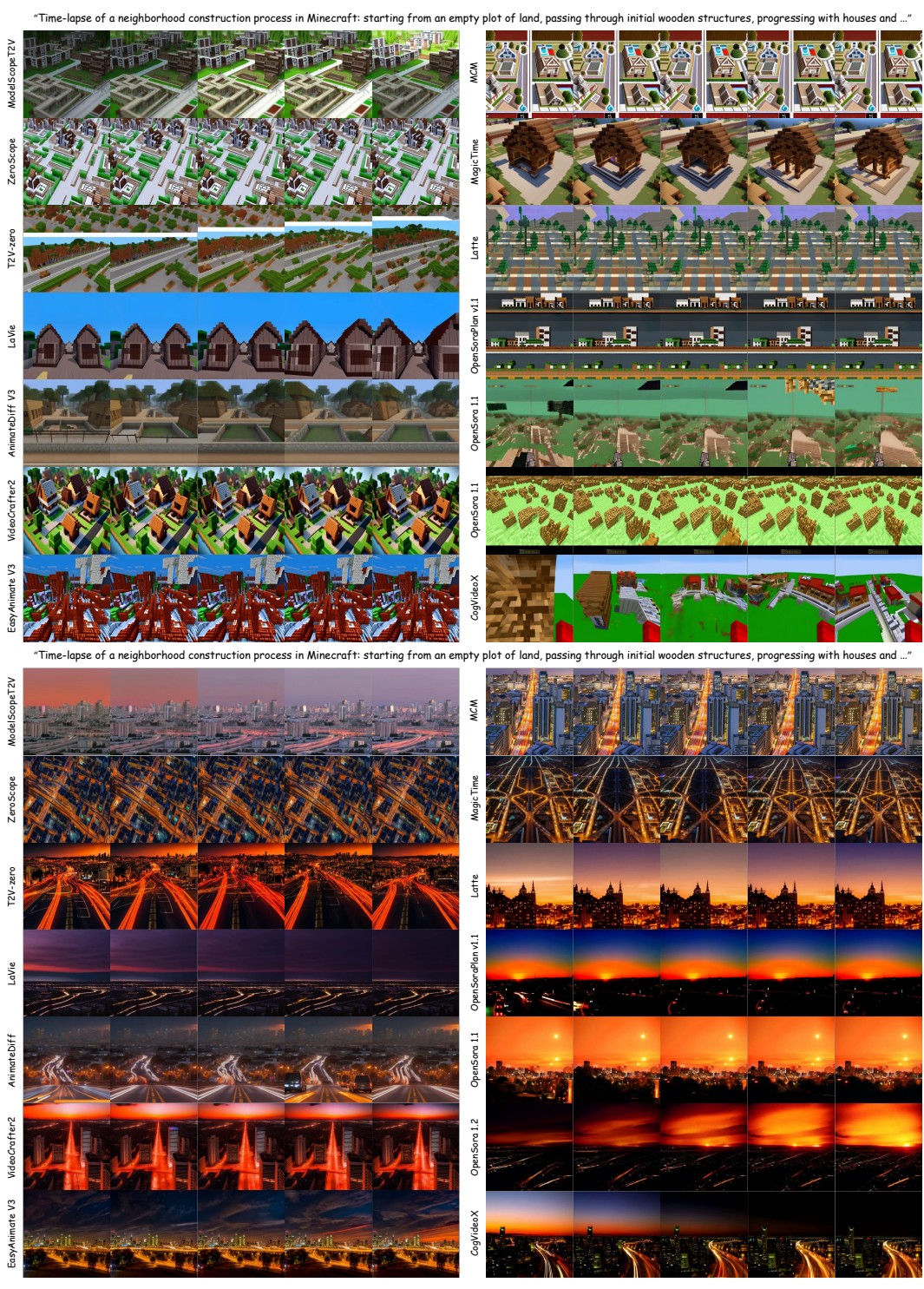

Figure 14: **More Qualitative Comparison with different T2V generation methods for the text-to-video task in ChronoMaigc-Bench.** Most methods struggle to follow the prompt to generate time-lapse videos with high physics prior content.

challenging full process of night-to-day transformation; no method can generate complex changes like plant growth or building construction. In terms of temporal coherence, the performance of

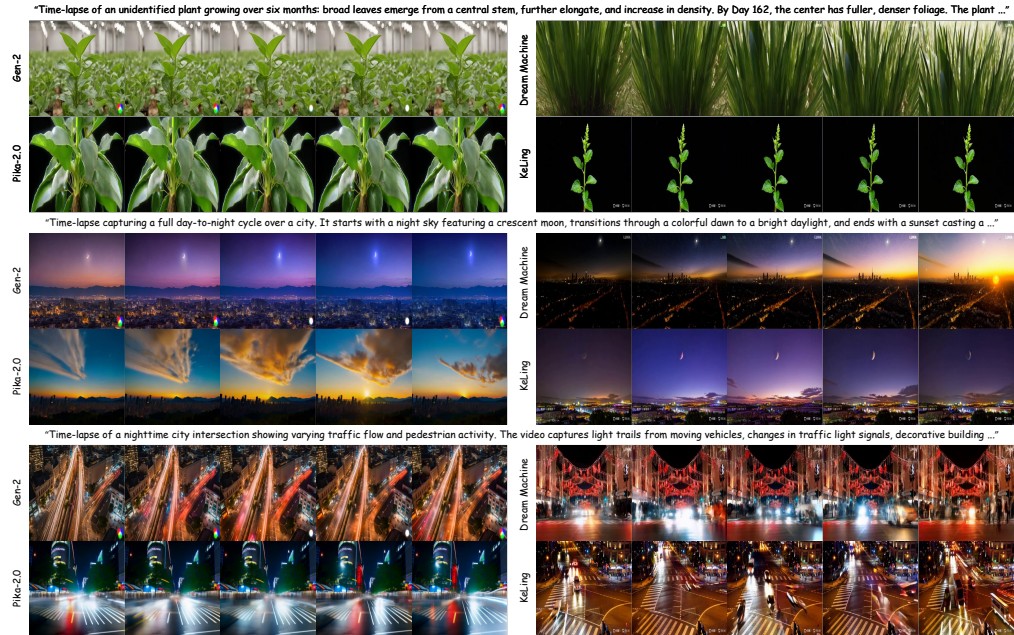

Figure 15: **Qualitative comparison with *Close-Source* generation methods for the text-to-video task in ChronoMaigc-Bench-150.** Most methods can only generate simple time-lapse videos such as traffic flows and starry skies, and are incapable of generating complex changes such as plant growth or building construction.

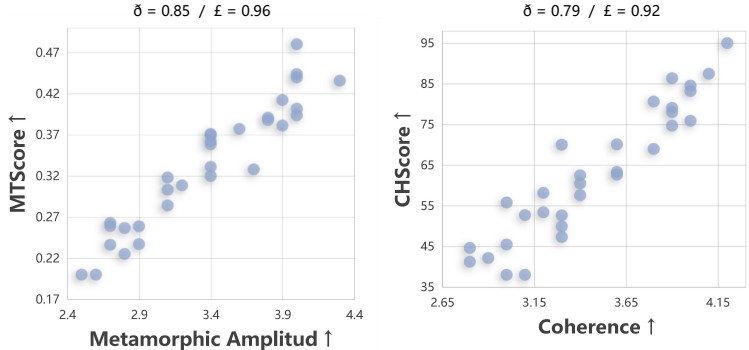

Figure 16: **Alignment between automatic metrics and human perception in terms of disaggregated data.** ð and £ represent Kendall↑ and Spearman↑ coefficients, respectively. ↑" denotes higher is better.

various closed-source models is comparable, with minor visible differences. Regarding visual quality, the DiT-based methods Dream Machine [48] and KeLing [35] outperform those based on U-Net, producing more realistic plants, more accurately saturated sky colors, and clearer traffic flow. In terms of text relevance, all methods adhere to the prompt's instructions to generate content relevant to the theme, except for Pika-1.0 [36], which mistakenly interprets day-to-night as night-to-day.

### D.6    Additional Details of Human Evaluation

**Pre-processing**    The questionnaire for human evaluators to rate the generated content was established following methodologies from prior studies [60, 72, 89, 66]. The evaluation focused on four primary aspects: *Visual Quality*, *Text Relevance*, *Metamorphic Amplitude*, and *Coherence*. For each criterion, we employed a five-point rating scale and provided scoring guidelines to ensure consistent user selections, thereby minimizing assessment bias. For detailed criteria, please refer to Figure 17. For detailed explanation of voters, the voter population predominantly comprises undergraduate, master's, and phd students from universities, along with a segment of the general public who are not

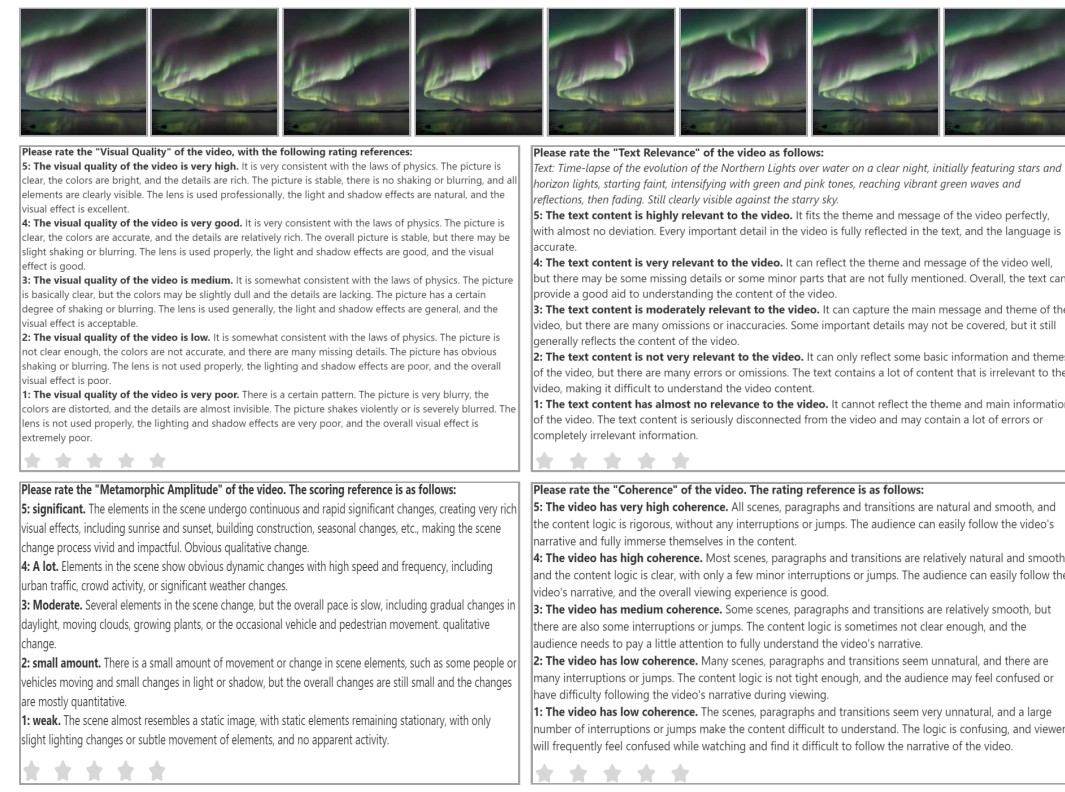

Figure 17: **Visualization of the Questionnaire for Human Evaluation.** We employ a five-point rating scale and provided scoring guidelines to ensure consistent selections by users, thereby minimizing assessment bias.

associated with this field. They come from various regions around the world, including China, USA, Singapore, etc., which ensures that the participants have universality. This composition guarantees the precision and diversity of human evaluations.

**Post-processing**    Given the use of a simple five-point evaluation scale, we remove outliers from the responses as follows:

- Restricted each IP address to prevent duplicates and required users to log in to their accounts before voting, ensuring that each person could only submit once.

- Determined the validity of data based on the time taken to complete the questionnaire. Given that completing a questionnaire typically takes 10 to 20 minutes, we excluded samples where the response time was less than 10 minutes.

- Randomized the order in which different videos were presented to avoid cognitive biases among voters.

- Required a sliding verification at submission to ensure that all questionnaires were completed manually and not by bots.

- Discarded any questionnaire where 50% of the ratings were extreme values, i.e., the sum of 5-point and 1-point options exceeded 50%.

**Additional Evaluation**    In addition to the main text, Figure 6 analyzes the video metrics aggregated by the model. We also provide a human evaluation of disaggregated data (that is, where each point represents a video), which consists of 32 videos randomly selected from all the questionnaires. The results are shown in Figure 16. It can be seen that the proposed MTScore and CHScore are consistent with human perception in terms of disaggregated data.

# E  More details about 75 subcategories in ChronoMaigc-Bench

Due to space limitations, we provide detailed descriptions of the 75 search terms used in ChronoMagic-Bench below (*each term includes the phrase "time-lapse"*), all of which pertain to time-lapse. Because of search engine limitations, some precise search terms may not yield optimal results. Therefore, to collect search terms more comprehensively, some overlap may exist between broader terms like "plant" and precise terms like "flower".

**Biological:**

- *Animal.* Captures the movements, behaviors, and interactions of various animals over an extended period. This includes everything from the daily activities of pets to the complex behaviors of wild animals in their natural habitats.

- *Spider Web.* Showcases the intricate process of spiders spinning their webs. It highlights the changes the web undergoes over time.

- *Butterfly.* Focuses on the life cycle of butterflies, particularly the metamorphosis from caterpillar to chrysalis to adult butterfly. It includes the intricate process of pupation and emergence.

- *Hatching.* Documents the hatching process of various eggs, including those of birds, reptiles, and insects. This category captures the moment of emergence and the initial activities of the newborns.

- *Flower Dying.* Captures the end-of-life process of flowers, showing how they wilt and decay over time.

- *Mealworm.* Showcases the behavior of mealworms, including their feeding habits.

- *Plant Growing.* This broad category includes time-lapse videos of various plants as they grow from seeds to mature plants. It encompasses root development, stem elongation, and the emergence of leaves and flowers.

- *Ripening.* Documents the ripening process of fruits and vegetables, showing the changes in color, texture, and overall appearance as they become ready for consumption.

- *Leaves.* Focuses on the growth, movement, and changes of leaves on plants. This includes the unfolding of new leaves, changes in color, and responses to environmental factors.

- *Seed.* Captures the germination and initial growth stages of seeds, from the first signs of sprouting to the establishment of seedlings. It focuses on the early and often delicate stages of plant development.

- *Blooming.* Showcases the process of flowers blooming, capturing the gradual opening of petals and the transformation from buds to full blossoms.

- *Mushroom.* Captures the rapid growth and development of mushrooms, from the initial emergence of the mycelium to the full development of the fruiting body.

**Human Creation:**

- *3D Printing.* Captures the process of 3D printing objects. These videos show the additive manufacturing process layer by layer, from the initial base to the final, complete object.

- *Painting.* Showcases the process of creating a painting, from the initial sketch to the final strokes.

- *Laser Engraving.* Show the process of laser engraving on various materials, such as the process of pattern formation.

- *Building.* Documents the construction of various structures, including residential, commercial, and industrial buildings. This category highlights the step-by-step development from foundation to completion.

- *Minecraft Build.* Captures the construction of complex structures and landscapes within the game Minecraft.

- *Demolition.* Captures the process of demolishing buildings and structures.

- *Fireworks.* Captures the display of fireworks, showcasing the entire process from the launch of the explosive into the sky to its transformation into bursts of color and patterns in the night sky.

- *People.* Focuses on the activities and movements of people in various settings, including streets, parks, and public spaces.

- *Sport.* Captures sporting events and activities, highlighting the movement of athletes, the progression of games, and the energy of the crowd.

- *City.* Focuses on the dynamic activities within a city, including urban development, traffic flow, and daily life. These videos often showcase the bustling and ever-changing nature of urban environments.

- *Factory.* Highlights the operations within a factory, including assembly lines, manufacturing processes, and the movement of goods.

- *Market.* Documents the activities within a market, including the setting up of stalls, movement of people, and trading of goods.

- *Office.* Captures the daily activities within an office environment, including the ebb and flow of workers, meetings, and the general hustle and bustle of office life.

- *Restaurant.* Documents the activities within a restaurant, including food preparation, service, and customer interactions.

- *Road.* Capture the traffic flow, and changes in road conditions over time.

- *Station.* Focuses on the activities within transportation stations, such as train stations, bus terminals, and airports. These videos capture the flow of passengers, arrivals, departures, and the hustle and bustle of travel hubs..

- *Traffic.* Captures the movement of vehicles on roads and highways, including the traffic flow, congestion, and the changing pace of vehicular movement throughout the day.

- *Walking.* Focuses on people walking in various environments, such as city streets, parks, and malls.

- *Parking.* Captures the movement of vehicles in parking lots or garages, including the flow of cars as they enter, park, and exit.

**Meteorological:**

- *Day to Night.* Show the transitions from daylight to nighttime, capturing the gradual shift in light and atmosphere as day turns to night.

- *Night to Day.* Shows the transitions from nighttime to daylight, showing the gradual change in lighting and environment as night turns to day.

- *Day.* Captures the progression of daylight hours, highlighting changes in light intensity, shadows, and weather conditions.

- *Night.* Shows the sequences of nighttime scenes, often capturing the movement of stars, phases of the moon, and nocturnal activities.

- *Cloud.* Shows the formation, movement, and dissipation of clouds, providing a dynamic view of the ever-changing sky.

- *Lunar Eclipse.* Shows the gradual movement of the moon through the Earth's shadow and the resulting changes in appearance during a lunar eclipse.

- *Rainbow.* Captures the formation, duration, and fading of rainbows, providing a colorful display over time.

- *Sky.* Captures a variety of atmospheric phenomena such as cloud movements, sunrises, sunsets, and weather changes over time.

- *Snowstorm.* Shows the accumulation of snow and the changing conditions during and after a snowstorm.

- *Storm.* Highlights the intensity and movement of storm clouds and lightning during various types of storms.

- *Sunrise.* Captures the gradual increase in light and the awakening of the environment during sunrise.

- *Sunset.* Showcases the beautiful colors and gradual fading of light as the day ends during sunset.

- *Aurora.* Captures the dynamic changes and movement of the Northern and Southern Lights, showcasing the evolving natural light displays over time.

- *Tide.* Illustrates the rise and fall of sea levels and their impact on coastal landscapes over time.

- *Wind.* Captures the effects of wind on landscapes, including the movement of vegetation, dust storms, and changing cloud patterns over time.

- *Seasons.* Shows the dramatic changes across different seasons, highlighting the transformation of landscapes throughout the year.

- *Nature.* Captures various natural scenes, including the growth of plants, changes in landscapes, and wildlife activity.

- *Beach.* Illustrate the changes in tides, waves, and shifting weather conditions throughout the day.

- *Desert.* Shows the dramatic changes in light, temperature, and atmosphere in desert landscapes over time.

- *Forest.* Illustrates changes in foliage, light patterns, and wildlife activity in forests throughout the day or seasons.

- *Grassland.* Highlight the subtle yet significant changes in vegetation and weather in grasslands over time.

- *Lake.* Captures reflections, water level changes, and the transformation of surrounding landscapes.

- *Mountain.* Showcases changes in light, weather, and cloud movement around mountainous peaks over time.

- *Ocean.* Highlights the continuous motion of waves, tides, and the impact of weather on ocean scenes over time.

- *Plain.* Shows the transformation of open landscapes due to changing light and weather conditions over time.

- *River.* Illustrates the flow of water, changes in water levels, and the transformation of surrounding landscapes over time.

- *Valley.* Highlights changes in light, weather, and seasonal transformations in valley areas over time.

**Physical:**

- *Baking.* Shows the transformation of dough or batter as it rises and turns into baked goods, highlighting changes in color, texture, and volume over time.

- *Cooking.* Shows the various stages of food preparation and cooking, highlighting changes in texture, color, and form.

- *Candle Burning.* Illustrates the gradual melting and burning of a candle, including changes in the wax and the flickering flame.

- *Tea Diffusing.* Illustrates how tea leaves release their color and flavor into hot water, showing the gradual diffusion process and changes in the liquid.

- *Corrosion.* Captures the slow process of materials deteriorating due to chemical reactions with their environment, often resulting in rust or other forms of decay.

- *Decompose.* Shows organic materials breaking down over time, illustrating the process of decomposition and the changes in form and structure.

- *Fruit Rotting.* Illustrates the gradual decay and breakdown of fruit, showing changes in color, texture, and structure as it rots.

- *Explosion.* Captures the rapid and dramatic release of energy, showing the sudden change in materials and the environment.

- *Burning.* Captures the process of combustion, showing how materials ignite, burn, and reduce to ash or other residues.

- *Gasification.* Shows the process of a solid or liquid turning into gas, highlighting the changes in state and movement of particles.

- *Ice Melting.* Captures the transition of ice from solid to liquid, showing the gradual melting process and changes in shape and volume.

- *Ink Diffusing.* Illustrates how ink spreads and disperses in a liquid, showing the dynamic patterns and changes in concentration over time.

- *Melting.* Shows the process of a solid turning into a liquid, highlighting changes in form and consistency as the material melts.

- *Rusting.* Captures the slow formation of rust on metal surfaces, showing the chemical changes and resulting texture and color changes.

- *Water Freezing.* Shows the transition of water from liquid to solid, capturing the formation of ice and changes in volume and structure.

## F Ethics Statement

**Potential Harms Caused by the Research Process.** The video data utilized by ChronoMagic-Bench is sourced from free content available on four platforms: Pexels (CC0), MixKit (CC0), PixaBay (CC0), and YouTube (CC BY 4.0). Conversely, ChronoMagic-Pro exclusively employs videos from YouTube (CC BY 4.0). The licensing types of these videos are clearly indicated on their respective platforms. The CC0 license (Creative Commons Zero) designates content as public domain, allowing unrestricted use without the need for additional permissions or licenses. Videos from the YouTube platform adhere to the CC BY 4.0 license (Creative Commons Attribution 4.0); consequently, we have included video IDs and author information in the metadata to prevent any potential contractual disputes. The video content consists entirely of time-lapse footage, and we detect and discard NSFW content based on the video caption. For videos involving identifiable individuals, we accelerate the blurring process to ensure the security of personally identifiable information. The collected videos are organized into four major categories (comprising 75 subcategories), with contributors hailing from various countries and regions worldwide. This diversity ensures that ChronoMagic-Bench and ChronoMagic-Pro possess ample representativeness. The Open-Sora-Plan model [43], fine-tuned using our dataset, exhibited no significant content bias.

Data collection was facilitated by the dedicated efforts of numerous contributors, including the authors of this paper and those who participated in the human evaluation. We regard an individual's hourly wage or compensation as personal information, which, due to privacy considerations, cannot be disclosed. Nonetheless, we can confirm that all participants received appropriate compensation in compliance with the legal requirements of their respective countries or regions. The privacy information of all participants is protected, so there is no additional risk to the them.

**Societal Impact and Potential Harmful Consequences.** The objective of ChronoMagic-Bench is to identify the limitations of current text-to-video generation models in producing time-lapse videos and to develop the ChronoMagic-Pro dataset to advance the field. Although time-lapse video generation models offer substantial potential to support and enhance human creativity, it is crucial to consider broader societal implications during their development:

First and foremost, environmental issues cannot be overlooked. As text-to-video generation technology advances, the demand for computational resources escalates. Large-scale data processing, model testing, and training generally depend on energy-intensive data centers, which significantly contribute to carbon emissions. For instance, this study utilized the energy-intensive H100 for experiments. If

not addressed, the widespread adoption of this technology could further exacerbate climate change. Consequently, researchers and developers should focus on optimizing algorithms to reduce energy consumption.

Secondly, the generation of false content has raised significant social and ethical concerns. After appropriate fine-tuning using the ChronoMagic-Pro/ProH dataset, text-to-video generation models are capable of producing not only metamorphic videos with extended time spans and high levels of realism but also high-quality general videos. These generative models could be misused to create deceptive videos, potentially misleading the public or disseminating misinformation, particularly on fast-paced and widely influential platforms such as social media. To prevent such misuse, it is crucial to consider the implementation of content authenticity verification mechanisms and the establishment of robust legal and ethical frameworks during the development and deployment of these technologies.

Lastly, the issue of dataset bias may result in skewed and inequitable outcomes in model generation. Although the video content in the ChronoMagic-Bench and ChronoMagic-Pro datasets is sourced globally, the captions are exclusively in English. This single-language choice may impair the model's ability to accurately interpret and generate videos across diverse cultural contexts and non-English language environments. Moreover, this bias could exacerbate existing language inequalities by disregarding the needs of non-English-speaking users. Therefore, future dataset construction should incorporate multilingual support to ensure broader adaptability and fairness in models on a global scale.

**Impact Mitigation Measures.** We take full responsibility for the licensing, distribution, and maintenance of our ChronoMaigc-Bench and ChronoMagic-Pro/ProH. Our datasets and benchmark are released under a CC-BY-4.0 license, and our code under an Apache license. We have clearly stated on our homepage that all data is for academic research only to prevent misuse or improper use. And we provide the email address for YouTube authors to contact and remove invalid videos in time. All the metadata are hosted on GitHub and HuggingFace at the following URLs: https://github.com/PKU-YuanGroup/ChronoMagic-Bench and https://huggingface.co/collections/BestWishYsh.

