# OpenReview forum: "ChronoMagic-Bench: A Benchmark for Metamorphic Evaluation of Text-to-Time-lapse Video Generation"
_NeurIPS.cc/2024/Datasets_and_Benchmarks_Track — NeurIPS 2024 Track Datasets and Benchmarks Spotlight_

### Official Review · Reviewer_ntFp · 2024-07-19
**Promising paper but could benefit from multiple improvements**

**Rating:** 7
**Confidence:** 4

**Review:**

**Significance**

I have some mixed thoughts about this. It's true that time lapse generation is a problem for many video generation models, and this work allows to measure this issue and to help alleviate this problem. By itself, however, I'm not sure this problem has enough wide significance for the field. But as I describe in one of the sections below, I think there's potential in using the provided dataset to improve on some aspects of general video generation as well.

**Originality**

As I understand, this is the first work attacking the problem of time lapse generation by using a dataset of this scale. The provided metrics appear novel too. Overall I think this is original work.

**Quality**

The dataset appears to be high quality, and the construction process is appropriate. As for the metrics, some details details regarding the comparison to human evaluation appear to be missing, which I think would help evaluate the metric quality. I also appreciated the experimental results of fine-tuning, which I think would elevate the quality of the paper if moved from the appendix to the main article.

There are also other significant issues or possible improvements I mention in the sections below, which currently decrease the quality of the paper.

**Clarity**

The paper is clear for the most part, but it would benefit from proof-reading as it has some amount of misspellings and minor redaction issues.

**Strengths:**

Multiple aspects of this paper are very interesting:

* The paper introduces a complete dataset of time lapse videos, of good enough size for benchmark and fine-tuning uses, and as complement of other datasets. Given the challenge it represents to build good video datasets, this effort is greatly appreciated.

* The metrics introduced allow measuring important but often overlooked aspects of video generation.

* The benchmark compares many video models with these and other metrics, producing interesting conclusions

* A method for comparing models based on these metrics is presented, which is novel and useful

* An experiment shows that fine-tuning a video model on this dataset elevates generation quality for time lapses

Overall, the paper displays appropriate research and engineering work, and it makes progress on solving the stated problem.

**Additional Feedback:**

I think this is good work but the presentation appears to me, to be below the standards of publication. I hope the authors can work towards improvement on the suggested issues and I could consider increasing the score in that case.

**Clarity:**

As I mentioned above in more detail, while the paper is clearly written, it would benefit from significant proof-reading.

**Correctness:**

Dataset construction looks correct. Metrics look appropriate as well. Experiment design and analysis appears correct.

**Documentation:**

Code and data is promised in the paper but not yet included. As a result, all of URL for reviewer access to the dataset, hosting, and maintenance plan are missing.

Dataset construction discussion is good.

Ethical discussion is missing.

**Ethics:**

Ethical discussion is missing from the paper. Because the topic involves video generation (for which a discussion on fake videos and other related issues is appropriate), video datasets (for which discussion on privacy, copyright and consent is granted), I recommend the authors add a discussion on ethics, and also think the publication would benefit from ethics review.

**Limitations:**

I suggest that the authors add a discussion about the ethics and societal impact (see details above)

**Opportunities For Improvement:**

Overall I think this paper is promising, but there are many limitations and opportunities of improvement which decrease the quality of the result:

* I personally find the experiment on the fine-tuned model very interesting, and I think making it more central by moving it from the appendix to the main article would elevate the quality of the publication.
  * It would also be very interesting to see how the fine-tuned model performs on prompts that aren't about time-lapses. Is it possible to fix the tendency of many video models to generate almost still videos for arbitrary topics, by fine-tuning on this dataset? If this was found to be true, it would make the paper much more widely relevant to the community.

* The answer to the question "*Did you discuss any potential negative societal impacts of your work?*" is yes, but there's no discussion on ethics on the paper nor appendix. It's worth discussing some aspects of video generation (fake content, copyrighted data in the dataset, for example)

* The answer to this question is also positive "*Did you include the code, data, and instructions needed to reproduce the main experimental results (either in the supplemental material or as a URL)*", but there's no data nor source available yet. When will this be included?

* Other questions are also answered "yes" but not really discussed (maybe for some, the answer is "not relevant" rather than "yes"?):

  * "*Did you discuss whether and how consent was obtained from people whose data you’re using/curating?*" - given the dataset is based on public data this is worth commenting on
  * "*Did you include the estimated hourly wage paid to participants and the total amount spent on participant compensation?*"
  * "*Did you include the total amount of compute and the type of resources used (e.g., type of GPUs, internal cluster, or cloud provider)?*" (type is included, but not amounts)

* Regarding ethics and privacy, it would be good to provide means for video authors to remove their content from the dataset

* Texts like "*A normal video, not a time-lapse video*" are used for MTScore. Does the template "*An x video, not a y video*" really work better than "An x video"? I could see some video models getting confused by the negative part of the phrase

* What are the 16 prompts used to do human preference and alignment evaluation? I'm especially interested in seeing whether these prompts show enough variety, and display cases of both time lapse and non-time lapse for a fair analysis

* All terms in the divisor of `CHScore` are weighted the same; are there opportunities for improving this metric by finding a best ratio for each term?

* Figure 6 analyzes the video metrics as aggregated by model. Can a similar figure be computed for disaggregated data (that is, where each point is a video)?

* Regarding "*Future efforts will focus on aligning automated metrics more closely with human judgment for a more accurate evaluation of T2V models*" (and figure 6), maybe worth expanding on what a potential direction is.

* It could be interesting to include non-generated videos in the human preference evaluation, especially if the results differ from those of generated videos. This could also be practical to do, because there's no inference cost

* In the discussion about fine-tuning in the appendix, it is mentioned that coherence remains uncompromised, which is contradicted by one of the following phrases (and by the table)

* [minor] It would be good to compare against closed models too, like Pika, Gen-2

* [minor] Could the `R_missed` formula be simplified by using `m[i]`?

* The paper has multiple typos like "*Base on ChronoMagic-Bench*", "*we conducte*", "*about ChronoMaigc-Pro*", "*ChronoMaigc-Bench*". Also some places may benefit from slight rewriting, e.g. I think the phrase "*Unlike before*" is redundant. Overall, the article would benefit from some proof-reading

**Relation To Prior Work:**

Prior art and its relation to this work is discussed in detail.

**Summary And Contributions:**

Generative video models tend to produce time lapses of lower quality than more general videos. This paper works on a solution to this problem based on producing a dataset of time lapse videos, and a benchmark for coherence and movement.

The authors generate the dataset by extracting video clips using a curated set of prompts, and some processing including scene change detection and quality filters. They also determine a set of metrics, build a benchmark on those, and compare the metrics against human evaluation. They also provide a guideline for model selection based on these metrics.

A complete appendix is included as well, with multiple details about the dataset and metrics, and extra fine-tuning experiments.

---

> ### Author Rebuttal · Authors · 2024-08-17
>
> Thank you for the time, thorough comments, and nice suggestions. We are pleased that you acknowledged the metrics, technical soundness, and effective experiments.
> >**Q1**: Experiment on the fine-tuned model is very interesting, and can make it more central by moving it from the appendix to the main article would elevate the quality of the publication.
>
> **A1**: Thanks for your suggestions. We will move it from the appendix to the main article in the latest version.
>
> >**Q2**: It would also be very interesting to see how the fine-tuned model performs on prompts that aren't about time-lapses. Is it possible to fix the tendency of many video models to generate almost still videos for arbitrary topics, by fine-tuning on this dataset? If this was found to be true, it would make the paper much more widely relevant to the community.
>
> **A2**: Yes, it can fix the tendency of many video models to generate almost still videos for arbitrary topics, if appropriately fine-tuned. This phenomenon has also been validated in *[MagicTime-DiT](https://github.com/PKU-YuanGroup/MagicTime)*, even though it used only over 2,000 time-lapse videos.
>
> >**Q3**: Add a discussion about the ethics and societal impact. It's worth discussing some aspects of video generation (fake content, copyrighted data in the dataset, for example)
>
> **A3**: Thanks for your suggestions. We will add the following content to the latest version:
> - After appropriate fine-tuning using the ChronoMagic-Pro/ProH dataset, text-to-video generation models are capable of producing not only metamorphic videos with extended time spans and high levels of realism but also high-quality general videos. These generative models could be misused to create deceptive videos, potentially misleading the public or disseminating misinformation, particularly on fast-paced and widely influential platforms such as social media. To prevent such misuse, it is crucial to consider the implementation of content authenticity verification mechanisms and the establishment of robust legal and ethical frameworks during the development and deployment of these technologies.
> - The videos utilized by ChronoMagic-Bench is sourced from free content available on four platforms: Pexels (CC0), MixKit (CC0), PixaBay (CC0), and YouTube (CC BY 4.0). Conversely, ChronoMagic-Pro exclusively employs videos from YouTube (CC BY 4.0). The licensing types of these videos are clearly indicated on their respective platforms. Our datasets and benchmark are released under a CC BY 4.0 license, and our code under an Apache license. We will state on our *[homepage](https://pku-yuangroup.github.io/ChronoMagic-Bench)* that all data is for academic research only to prevent misuse or improper use.
> - Due to limited space, we will also add discussions on environmental pollution and data bias in the latest version.
>
> >**Q4**: The answer to this question is also positive "Did you include the code, data, and instructions needed to reproduce the main experimental results (either in the supplemental material or as a URL)", but there's no data nor source available yet. When will this be included?
>
> **A4**: Sorry for the confusion, we have uploaded all the relevant content to *[Page](https://pku-yuangroup.github.io/ChronoMagic-Bench)*, *[Code](https://github.com/PKU-YuanGroup/ChronoMagic-Bench)* and *[Dataset & Benchmark](https://huggingface.co/collections/BestWishYsh/chronomagic-bench-667bea7abfe251ebedd5b8dd)*.
>
> >**Q5**: Other questions are also answered "yes" but not really discussed (maybe for some, the answer is "not relevant" rather than "yes"?):
> >- "Did you discuss whether and how consent was obtained from people whose data you’re using/curating?" - given the dataset is based on public data this is worth commenting on
> >- "Did you include the estimated hourly wage paid to participants and the total amount spent on participant compensation?"
> >- "Did you include the total amount of compute and the type of resources used (e.g., type of GPUs, internal cluster, or cloud provider)?" (type is included, but not amounts)
>
> **A5**: Thanks for your detailed review. We will make the following additions in the latest version:
> - ChronoMagic-Bench sources video data from Pexels, MixKit, PixaBay (all CC0), and YouTube (CC BY 4.0), while ChronoMagic-Pro uses only YouTube (CC BY 4.0) videos. CC0 content is public domain, requiring no additional permissions. For YouTube videos under CC BY 4.0, we include video IDs and author information in the metadata to avoid contractual disputes. We declare that the dataset is for academic use only and cannot be used for commercial purposes.
> - We treat an individual’s hourly wage or compensation as confidential personal information, which cannot be shared due to privacy concerns. However, we can assure that all participants were fairly compensated in line with the legal standards of their respective countries or regions.
> - We used 8x H100 to obtain all inference results of different video generation models and then calculated the metrics for the generated videos on 4x A800.
>
> >**Q6**: Regarding ethics and privacy, it would be good to provide means for video authors to remove their content from the dataset
>
> **A6**: Thanks for your constructive suggestion. We have provided a dedicated contact email in [GitHub](https://github.com/PKU-YuanGroup/ChronoMagic-Bench) for video creators to request content removal, and they can also directly submit an issue to request the removal of video content.

---

> ### Author Rebuttal · Authors · 2024-08-17
>
> >**Q7**: Texts like "A normal video, not a time-lapse video" are used for MTScore. Does the template "An x video, not a y video" really work better than "An x video"? I could see some video models getting confused by the negative part of the phrase.
>
> **A7**: Yes, for *[InternVideo2](https://github.com/OpenGVLab/InternVideo/tree/main/InternVideo2)*, we found that the simple phrase "An x video" could not consistently produce a stable score, no matter how the description was modified. Only when using "An x video, not a y video" could the model output stable and accurate results.
>
> >**Q8**: What are the 16 prompts used to do human preference and alignment evaluation? I'm especially interested in seeing whether these prompts show enough variety, and display cases of both time lapse and non-time lapse for a fair analysis
>
> **A8**: We have uploaded the prompts to this links: *[view all prompts](https://github.com/user-attachments/files/16634768/human_evaluation_prompts.csv)*. The prompts are randomly selected from four principal time-lapse categories, with each category represented by four distinct sub-category prompts, thereby ensuring a diverse range of subjects. Given our focus on assessing the model's proficiency in generating time-lapse videos, the selection excludes non-time lapse prompts.
>
> >**Q9**: All terms in the divisor of CHScore are weighted the same; are there opportunities for improving this metric by finding a best ratio for each term?
>
> **A9**: Thanks for your comment. We improve the accuracy of CHScore by standardizing each term to a range of 0 to 1 and setting the ratio for each term, and the latest results can be accessed on the *[LeadBoard](https://huggingface.co/spaces/BestWishYsh/ChronoMagic-Bench)*. Specifically, *R_missed* is a global metric that reflects the overall performance of the model across the entire video, and it is the most important, with a weight of 0.35; *V_missed* reflects the stability of the variation in the proportion of missed points between frames. Stability is crucial in video analysis, so it is assigned a weight of 0.25; *R_cut* is an indicator that reflects abnormal situations, with a weight of 0.15; *C_missed* serves a similar function to the *R_cut* and can be considered a secondary indicator, with a weight of 0.15; *M_missed* focuses on individual extreme cases, so it can be given a lower weight, with a weight of 0.10. The latest *[human evaluation](https://pku-yuangroup.github.io/ChronoMagic-Bench/static/images/human_evaluation.jpg)* shows that the improved CHScore is more in line with human perception.
>
> |  | ModelScopeT2V | ZeroScope | T2V-zero | LaVie | AnimateDiff | VideoCrafter2 | MagicTime | Latte | OpenSora 1.1 | OpenSoraPlan v1.1 |
> |---------|---------------|-----------|----------|-------|-------------|---------------|-----------|-------|--------------|-------------------|
> | **CHScore**                    | 11.03 | 25.13 | 1.68 | 8.60 | 11.36 | 8.27 | 10.66 | 13.81 | 10.03 | 10.35 |
> | **CHScore (+direction & weight)** | 29.23 | 46.13 | 8.62 | 28.01 | 30.35 | 27.80 | 29.03 | 34.73 | 25.34 | 23.15 |
>
> >**Q10**: Figure 6 analyzes the video metrics as aggregated by model. Can a similar figure be computed for disaggregated data (that is, where each point is a video)?
>
> **A10**: Thanks for your suggestion, we visualized the results from one of the questionnaires, which included data from 32 generated videos. As illustrated in *[Figure](https://github.com/user-attachments/assets/61bfd671-4337-4b2c-be40-c456607faa7f)*, the resulst reaffirm the validity of the proposed metrics.
>
> >**Q11**: Regarding "Future efforts will focus on aligning automated metrics more closely with human judgment for a more accurate evaluation of T2V models" (and figure 6), maybe worth expanding on what a potential direction is.
>
> **A11**: Thanks for your comments. One potential future direction involves transforming the scoring process into a question-and-answer format using Multimodal Large Language Models (MLLMs), a method that more closely aligns with human evaluation. In contrast, most existing automated metrics depend on model feature-based mapping, which tends to have lower interpretability and robustness. We have explored this direction with GPT4o (GPT4o-MTScore); however, due to the high costs of closed-source models and the limitations of open-source models, we have not yet broadly implemented this method.
>
> >**Q12**: It could be interesting to include non-generated videos in the human preference evaluation, especially if the results differ from those of generated videos. This could also be practical to do, because there's no inference cost.
>
> **A12**: Thanks for your constructive suggestions. The additional results, as detailed on the *[Page](https://pku-yuangroup.github.io/ChronoMagic-Bench/)* titled '*[Validation of the Automatic Metrics](https://pku-yuangroup.github.io/ChronoMagic-Bench/static/images/human_evaluation.jpg)*,' indicate that the scores of non-generated videos significantly surpass those of the generated videos.

---

> ### Author Rebuttal · Authors · 2024-08-17
>
> >**Q13**: In the discussion about fine-tuning in the appendix, it is mentioned that coherence remains uncompromised, which is contradicted by one of the following phrases (and by the table)
>
> **A13**: First, due to limited computational resources, we only selected 10,000 video-text pairs from ChronoMagic-Pro for training, and the training did not fully converge. Second, we initially employed a basic method to fine-tune the model, with the primary goal of verifying the dataset's impact on enhancing metamorphic amplitude. However, this simplistic fine-tuning strategy may result in flickering in the generated videos, manifesting as a decrease in CHScore—a phenomenon confirmed in the [MagicTime](https://github.com/PKU-YuanGroup/MagicTime). To address this issue, we adopt MagicTime's training strategy to fine-tune the [OpenSoraPlan v1.1](https://github.com/PKU-YuanGroup/Open-Sora-Plan) model. The results (did not fully converge), as presented in the table below, demonstrate improvements across all metrics.
>
> | **Method**                                      | **Venue**  | **UMT-FVD ↓** | **UMTScore ↑** | **MTScore ↑** | **CHScore ↑** | **GPT4o-MTScore ↑** |
> |-------------------------------------------------|------------|---------------|----------------|---------------|---------------|----------------------|
> | OpenSoraPlan v1.1                         | Github'24  | 188.53        | 2.421          | 0.327         | 23.15     | 2.19                 |
> | OpenSoraPlan v1.1 + ChronoMagic-Pro 10K + simple finetune       | ours        | 185.72    | 2.753      | 0.341     | 15.62         | 3.03             |
> | OpenSoraPlan v1.1 + ChronoMagic-Pro 10K + magictime finetune       | ours        | **180.11**    | **2.864**  | **0.346**  |  **25.72**  | **3.05**             |
>
>
> >**Q14**: It would be good to compare against closed models too, like Pika, Gen-2
>
> **A14**: Thank you for your suggestion. We conducted additional tests on four close-sourced models—Pika, Gen-2, KeLing, and LUMA as below. Due to our inability to perform large-scale pattern reasoning programmatically, we manually entered and tested 150 prompts from the ChronoMagic-Bench. Detailed results can be accessed on the *[LeadBoard](https://huggingface.co/spaces/BestWishYsh/ChronoMagic-Bench)*.
>
> | Model                                                    | Backbone | UMT-FVD↓ | UMTScore↑ | MTScore↑ | CHScore↑ | GPT4o-MTScore↑ |
> |----------------------------------------------------------|----------|----------|-----------|----------|----------|----------------|
> | [Gen-2 (20240610)](https://research.runwayml.com/gen2)   | DiT      | 218.99   | 2.400       | 0.373    | 70.95    | 2.62           |
> | [Pika-2.0 (20240610)](https://www.pika.art/)             | DiT      | 223.05   | 2.317     | 0.347    | 52.15    | 2.48           |
> | [Dream Machine (20240610)](https://lumalabs.ai/dream-machine) | DiT  | 214.91   | 2.387     | 0.474    | 44.04    | 3.11           |
> | [KeLing (20240610)](https://h5.kwaiying.com/officialWebsite) | DiT  | 202.32   | 2.517     | 0.369    | 62.84    | 2.74           |
>
>
> >**Q15**: Could the R_missed formula be simplified by using m[i]?
>
> **A15**: Thank you for the suggestion. We will revise the formula in the latest version to ensure clarity.
>
> >**Q16**: The paper has multiple typos. Overall, the article would benefit from some proof-reading.
>
> **A16**: Thank you very much! We will polish this paper and fix the grammatical errors.

---

> > ### Comment · Reviewer_ntFp · 2024-08-20
> >
> > Thank you for your replies, which answer to my questions in full detail. I'll be glad to revise my rating once a new version of the paper is available for review, please let me know.

---

> > > ### Comment · Reviewer_ntFp · 2024-08-22
> > >
> > > Thank you again for all the detailed replies. These answers satisfy my questions; I have updated my rating under the assumption that the discussed revisions will be applied to the camera ready paper. Thank you

---

> > > > ### Author Rebuttal · Authors · 2024-08-22
> > > >
> > > > Thank you for the quick reply and nice suggestions! We promise to apply all the discussed revisions to the final version of this paper (camera-ready).

---

### Official Review · Reviewer_qu3H · 2024-07-23

**Rating:** 8
**Confidence:** 5

**Review:**

**Quality**: The paper is well-structured, with a clear presentation of the problem, proposed solutions, and results. The quality of the research is high, as evidenced by the thorough evaluation of existing benchmarks and the introduction of new metrics and a dataset.

**Clarity**: The paper is written in a clear and concise manner. The figures and tables are well-designed and support the text effectively. The explanations of the new metrics and the dataset are particularly clear, However, the new indicators have only undergone minimal manual validation which need to be improved.

**Originality**: The work is highly original, addressing a gap in the evaluation of T2V models for time-lapse video generation. The introduction of MTScore and CHScore, as well as the ChronoMagic-Pro dataset, are significant contributions to the field.

**Significance**: Although the benchmark designed in this article is limited to time-lapse video, The benchmark and dataset have the potential to significantly impact the development of T2V models, providing a more comprehensive evaluation framework. The focus on metamorphic amplitude and temporal coherence is particularly important for advancing the state-of-the-art in video generation.

**Pros**:

1. **Innovative Benchmark**: The introduction of ChronoMagic-Bench fills a gap in the evaluation of T2V models for time-lapse video generation.
2. **New Metrics**: MTScore and CHScore are well-designed and align with human perception, providing a more accurate assessment of video quality.
3. **Large-Scale Dataset**: ChronoMagic-Pro is a valuable resource for the research community, offering a diverse set of time-lapse videos.
4. **Comprehensive Evaluation**: The paper includes a thorough evaluation of existing open-source models, providing insights into their strengths and weaknesses and pointed out the potential improvement directions of these models.

**Cons**:

1. **Limited Model Scope**: The evaluation is currently restricted to open-source models, which may not represent the full range of T2V models available.
2. **Metric Validation**: While MTScore and CHScore show promise, further validation across a broader set of models and videos would strengthen their credibility,and these two indicators require more theoretical evidence to prove their effectiveness.
3. **Data Diversity**: Although the dataset is large, there may be a need for even greater diversity in terms of video content and scenarios to fully represent the complexity of real-world videos, eg. low pixel videos.

**Strengths:**

- **Innovative Benchmark**: The introduction of ChronoMagic-Bench fills a gap in the evaluation of T2V models for time-lapse video generation.
- **New Metrics**: MTScore and CHScore are well-designed and align with human perception, providing a more accurate assessment of video quality.
- **Large-Scale Dataset**: ChronoMagic-Pro is a valuable resource for the research community, offering a diverse set of time-lapse videos.
- **Comprehensive Evaluation**: The paper includes a thorough evaluation of existing open-source models, providing insights into their strengths and weaknesses and pointed out the potential improvement directions of these models.

**Additional Feedback:**

- Accessibility of the dataset: It is recommended to provide public access to the dataset for a wider research community to use.

- Reproducibility of the model: Providing implementation details and code of the model can help other researchers reproduce and validate the results.

- There is no clear explanation in the article why using GPT-4O as the evaluation criterion is feasible,it needs more detail and I think the number of people manually confirming the validity is relatively small, and there is no introduction on whether the participants have universality.

- The relevant experimental data is not open source, making subsequent work inconvenient.

**Clarity:**

The paper is written in a clear and concise manner. The figures and tables are well-designed and support the text effectively. The explanations of the new metrics and the dataset are particularly clear. Below are some advices:

- The layout in section 5.2 is not aesthetically pleasing, it can be aligned or wrapped.
- There are many redundant and repetitive statements in the article, such as explanations and explanations of the four evaluation indicators.

**Correctness:**

The research method and results of this article are logically correct, and the construction of evaluation indicators and datasets is also in line with the research objectives, the evaluation methods and experiment design appropriate and performed correctly,but a question need to ask,At present, most open source models have not shown good performance on the two indicators of new design. How to prove that these two indicators themselves are not unreasonable.

**Documentation:**

The article provides detailed experimental settings and evaluation methods, which can help other researchers reproduce and validate the results, But some of the content mentioned in the existing paper is in the appendix, which is not visible in the existing article.

**Ethics:**

The authors have made efforts to address ethical concerns in their submission, but a few areas warrant further discussion and review:

- The authors discuss how they obtained consent for the data they are using or curating and whether it contains personally identifiable information or offensive content. However, it would be beneficial to provide more detailed information on how consent was obtained and the specific measures taken to ensure data privacy and copyright compliance.
- There is no explicit discussion on how potential biases in the dataset and models are identified and mitigated and the benchmark evaluation results are not concrete enough to assess ethical issues.

**Limitations:**

The authors have addressed the limitations and potential negative societal impact of their work to some extent. However, there are areas where their discussion could be expanded for greater comprehensiveness and clarity. Here is a summary and some suggestions for improvement:

- More detailed explanations are needed regarding the evaluation data, and it is currently impossible to determine whether there is bias in the existing content.
- Expand the evaluation to include a broader range of models, including closed-source and proprietary models, to provide a more comprehensive assessment of the state-of-the-art in T2V generation.

**Opportunities For Improvement:**

- **Limited Model Scope**: The evaluation is currently restricted to open-source models, which may not represent the full range of T2V models available.
- **Metric Validation**: While MTScore and CHScore show promise, further validation across a broader set of models and videos would strengthen their credibility,and these two indicators require more theoretical evidence to prove their effectiveness.
- **Data Diversity**: Although the dataset is large, there may be a need for even greater diversity in terms of video content and scenarios to fully represent the complexity of real-world videos, eg. low pixel videos.

**Relation To Prior Work:**

This article proposes new benchmark testing and evaluation indicators based on existing research, which have clear correlations and comparisons with previous work. However, the article almost counted all the benchmarks of previous work, but the references did not mention some of the benchmarks for time-lapse videos, which raises doubts about the effectiveness of the benchmarks proposed by the author.

**Summary And Contributions:**

he paper introduces ChronoMagic-Bench, a novel benchmark for evaluating the capabilities of text-to-video (T2V) models in generating time-lapse videos. The benchmark focuses on the models' ability to produce videos with significant metamorphic amplitude and temporal coherence. Key contributions include:

- A new benchmark tailored for time-lapse video generation.
- Two new automatic metrics: MTScore for metamorphic amplitude and CHScore for temporal coherence.
- The creation of ChronoMagic-Pro, a large-scale dataset with 460k high-quality time-lapse video pairs and detailed captions.

---

> ### Author Rebuttal · Authors · 2024-08-17
>
> Thanks for your time and the constructive suggestions. Your recognition of the paper's structure and the rigorousness of our research methodologies is greatly appreciated. Here are additional responses and clarifications based on your comments.
>
> >**Q1**: Limited Model Scope: The evaluation is currently restricted to open-source models, which may not represent the full range of T2V models available.
>
> **A1**: In the latest version, we conducted additional tests on Gen-2, Pika, KeLing, and LUMA. The results are presented below. Further details can be found in our *[Leaderboard](https://huggingface.co/spaces/BestWishYsh/ChronoMagic-Bench)*.
>
> | Model                                                    | Backbone | UMT-FVD↓ | UMTScore↑ | MTScore↑ | CHScore↑ | GPT4o-MTScore↑ |
> |----------------------------------------------------------|----------|----------|-----------|----------|----------|----------------|
> | [Gen-2 (20240610)](https://research.runwayml.com/gen2)   | DiT      | 218.99   | 2.4       | 0.373    | 70.95    | 2.62           |
> | [Pika-2.0 (20240610)](https://www.pika.art/)             | DiT      | 223.05   | 2.317     | 0.347    | 52.15    | 2.48           |
> | [Dream Machine (20240610)](https://lumalabs.ai/dream-machine) | DiT  | 214.91   | 2.387     | 0.474    | 44.04    | 3.11           |
> | [KeLing (20240610)](https://h5.kwaiying.com/officialWebsite) | DiT  | 202.32   | 2.517     | 0.369    | 62.84    | 2.74           |
>
> >**Q2**: Metric Validation: While MTScore and CHScore show promise, further validation across a broader set of models and videos would strengthen their credibility,and these two indicators require more theoretical evidence to prove their effectiveness. (A question need to ask, At present, most open source models have not shown good performance on the two indicators of new design. How to prove that these two indicators themselves are not unreasonable.)
>
> **A2**: We utilize human evaluation, quantitative analysis, and qualitative analysis for cross-validation, thereby demonstrating the effectiveness of the proposed metrics. Furthermore, we have augmented the latest version with the following experiments:
>
> - An additional 41 participants were recruited to perform a human evaluation of OpenSora 1.2 and the Reference Videos. The results of eliminating outliers are shown in the *[Page-Validation of the Automatic Metrics](https://pku-yuangroup.github.io/ChronoMagic-Bench)*.
> - We performed a [qualitative analysis](https://github.com/user-attachments/assets/61bfd671-4337-4b2c-be40-c456607faa7f), akin to Figure 6, on the disaggregated data—where each data point represents a video—to validate the effectiveness of the proposed metrics.
> - We evaluated the closed-source models Gen-2, Pika, KeLing, and LUMA, alongside the latest open-source models OpenSora 1.2, OpenSoraPlan v1.2, EasyAnimate-V3, CogVideX, and MCM-MSLAION. The results are presented below. For a comprehensive overview of the results, please refer to the *[LeadBoard](https://huggingface.co/spaces/BestWishYsh/ChronoMagic-Bench)*.
>
> | Model | Backbone | UMT-FVD↓ | UMTScore↑	 | MTScore↑	 | CHScore↑	 | GPT4o-MTScore↑ |
> |-------|--------------|----------|----------|----------|----------|----------|
> | [OpenSora 1.2](https://github.com/hpcaitech/Open-Sora) | DiT | 210.93   | 2.681    | 0.383    | 15.71    | 2.5      |
> | [OpenSoraPlan v1.2](https://github.com/PKU-YuanGroup/Open-Sora-Plan) | DiT | 216.9    | 0.972    | 0.324    | 16.57    | 1.08     |
> | [EasyAnimate-V3](https://github.com/aigc-apps/EasyAnimate) | DiT | 202.03   | 2.733    | 0.352    | 20.52    | 2.33     |
> | [CogVideoX-2B](https://github.com/THUDM/CogVideo) | DiT | 195.52   | 3.24     | 0.472    | 35.38    | 3.09     |
> | [MCM-MSLION](https://yhzhai.github.io/mcm/) | U-NeT | 244.49 | 2.282 | 0.422 | 14.14 | 3.06
>
> >**Q3**: Data Diversity: Although the dataset is large, there may be a need for even greater diversity in terms of video content and scenarios to fully represent the complexity of real-world videos, eg. low pixel videos.
>
> **A3**: Thank you for your suggestion, we will collect more videos in the future. The ChronoMagic-Pro dataset comprises videos classified into four major categories and 75 subcategories, encompassing a wide range of processes from natural (e.g., blooming) to man-made (e.g., minecraft) phenomena. Additionally, we retained 13% of the videos with low aesthetic scores, which correspond to low pixel videos, to ensure sufficient diversity and to fully capture the complexity of real-world video content.
>
>
> >**Q4**: (1) More detailed explanations are needed regarding the evaluation data, and it is currently impossible to determine whether there is bias in the existing content.
> (2) Expand the evaluation to include a broader range of models, including closed-source and proprietary models, to provide a more comprehensive assessment of the state-of-the-art in T2V generation.
>
> **A4**: Thanks for the thorough comments. We make the following clarifications:
>
> - The evaluation data includes time-lapse video data spanning 4 major categories and 75 subcategories, ensuring adequate diversity. However, the prompts are provided exclusively in English. This single-language limitation may hinder the model's ability to accurately interpret and generate videos across diverse cultural contexts and non-English language settings. All metadata is accessible on *[HuggingFace](https://huggingface.co/collections/BestWishYsh/chronomagic-bench-667bea7abfe251ebedd5b8dd)*.
> - We have expanded the evaluation, please refer to the answer to **Q2**.
>
> >**Q5**: (1) The layout in section 5.2 is not aesthetically pleasing, it can be aligned or wrapped. (2) There are many redundant and repetitive statements in the article, such as explanations and explanations of the four evaluation indicators.
>
> **A5**: Thank you so much for your detailed review! We will polish this paper and fix these errors.

---

> ### Author Rebuttal · Authors · 2024-08-17
>
> >**Q6**: The article almost counted all the benchmarks of previous work, but the references did not mention some of the benchmarks for time-lapse videos, which raises doubts about the effectiveness of the benchmarks proposed by the author.
>
> **A6**: Sorry for the confusion. We will add all references in the latest version.
> - [Time-Lapse-D]: Xiong W, Luo W, Ma L, et al. Learning to generate time-lapse videos using multi-stage dynamic generative adversarial networks[C]//Proceedings of the IEEE Conference on Computer Vision and Pattern Recognition. 2018: 2364-2373.
> - [Sky Time-Lapse]: Xue H, Liu B, Yang H, et al. Learning fine-grained motion embedding for landscape animation[C]//Proceedings of the 29th ACM International Conference on Multimedia. 2021: 291-299.
> - [ChronoMagic]: Yuan S, Huang J, Shi Y, et al. MagicTime: Time-lapse Video Generation Models as Metamorphic Simulators[J]. arXiv preprint arXiv:2404.05014, 2024.
>
> >**Q7**: The article provides detailed experimental settings and evaluation methods, which can help other researchers reproduce and validate the results, But some of the content mentioned in the existing paper is in the appendix, which is not visible in the existing article.
>
> **A7**: Thanks for your constructive suggestion. We will move some useful content from the appendix to the main body in the latest version.
>
> >**Q8**: (1) The authors discuss how they obtained consent for the data they are using or curating and whether it contains personally identifiable information or offensive content. However, it would be beneficial to provide more detailed information on how consent was obtained and the specific measures taken to ensure data privacy and copyright compliance. (2) There is no explicit discussion on how potential biases in the dataset and models are identified and mitigated and the benchmark evaluation results are not concrete enough to assess ethical issues.
>
> **A8**: Thanks for your comment. We will make the following clarification in the latest version:
> - Most of the source videos are licensed under CC BY 4.0, with some under CC0. Since CC0 entails no copyright restrictions, no issues arise. For videos under CC BY 4.0, we include video IDs and author information in the metadata to prevent potential contractual disputes. Furthermore, we provide a contact email within the *[repository](https://github.com/PKU-YuanGroup/ChronoMagic-Bench)* to allow original video authors to request the prompt removal of any infringing content.
> - We will add the following in the latest version: "The video content consists entirely of time-lapse footage, and we detect and discard NSFW content based on the video caption. For videos involving identifiable individuals, we accelerate the blurring process to ensure the security of personally identifiable information. The collected videos are organized into four major categories (comprising 75 subcategories), with contributors hailing from various countries and regions worldwide. This diversity ensures that ChronoMagic-Bench and ChronoMagic-Pro possess ample representativeness. The Open-Sora-Plan model, fine-tuned using our dataset, exhibited no significant content bias."
>
> >**Q9**: Accessibility of the dataset: It is recommended to provide public access to the dataset for a wider research community to use.
>
> **A9**: Thanks for your advice. We have uploaded the dataset to *[HuggingFace](https://huggingface.co/collections/BestWishYsh/chronomagic-bench-667bea7abfe251ebedd5b8dd)*.
>
> >**Q10**: Reproducibility of the model: Providing implementation details and code of the model can help other researchers reproduce and validate the results.
>
> **A10**: Thanks for your suggestion. We have made the *[Code](https://github.com/PKU-YuanGroup/ChronoMagic-Bench)* publicly available.
>
> >**Q11**: There is no clear explanation in the article why using GPT-4o as the evaluation criterion is feasible,it needs more detail and I think the number of people manually confirming the validity is relatively small, and there is no introduction on whether the participants have universality.
>
> **A11**: For the first question, GPT-4o is currently the most human-aligned Multilingual Language Model (MLLM) available; we assume its results align with human perception and use it to validate the effectiveness of the low-cost MTScore.
> For the second question, the number of 171 voters is appropriate according to references [2][3][4][5] and far exceeds the 3 individuals mentioned in [1]. Additionally, we invited an additional 41 individuals for *[human evaluation](https://pku-yuangroup.github.io/ChronoMagic-Bench/static/images/human_evaluation.jpg)*, further substantiating the efficacy of the proposed metric.
> For the last question, the participants primarily comprised undergraduate, master’s, and phd students specializing in computer science, as well as a subset of individuals without prior exposure to this field. Participants were sourced globally, including representatives from China, USA, Singapore, and other regions, ensuring a broad and representative sample.
>
>
> [1]Liu Y, Li L, Ren S, et al. Fetv: A benchmark for fine-grained evaluation of open-domain text-to-video generation[J]. Advances in Neural Information Processing Systems, 2024, 36.
>
> [2]Huang Z, He Y, Yu J, et al. Vbench: Comprehensive benchmark suite for video generative models[C]//Proceedings of the IEEE/CVF Conference on Computer Vision and Pattern Recognition. 2024: 21807-21818.
>
> [3]Wu J Z, Fang G, Wu H, et al. Towards a better metric for text-to-video generation[J]. arXiv preprint arXiv:2401.07781, 2024.
>
> [4]Sun K, Huang K, Liu X, et al. T2V-CompBench: A Comprehensive Benchmark for Compositional Text-to-video Generation[J]. arXiv preprint arXiv:2407.14505, 2024.
>
> [5]Singer U, Polyak A, Hayes T, et al. Make-A-Video: Text-to-Video Generation without Text-Video Data[C]//The Eleventh International Conference on Learning Representations.

---

> > ### Author Rebuttal · Authors · 2024-08-17
> >
> > >**Q12**: The relevant experimental data is not open source, making subsequent work inconvenient.
> >
> > **A12**: Sorry for the inconvenience, we have uploaded all the relevant content to *[Page](https://pku-yuangroup.github.io/ChronoMagic-Bench)* and *[Dataset & Benchmark](https://huggingface.co/collections/BestWishYsh/chronomagic-bench-667bea7abfe251ebedd5b8dd)*.

---

### Official Review · Reviewer_4KT6 · 2024-07-26
**Time-lapse video benchmark and dataset**

**Rating:** 8
**Confidence:** 4
**Clarity:** Yes

**Review:**

- The paper is clearly written and organized.
- It addresses a gap in the evaluation of T2V models with absence of benchmarks for timelapse videos. Futher, introduction of two automated metrics grounded on specifically for timelapse videos make it useful for the community.

**Strengths:**

- Focus on time-lapse videos and metamorphic attributes fills a gap in existing T2V benchmarks.

- Both MTScore and CHScore are detailed metrics for time-lapse videos and have high correlation with human perception.

- Identified cases where existing T2V models fails to generate videos, like seed germination, egg hatching, or sunrise.

**Additional Feedback:**

No

**Correctness:**

The claims made in the submission appear to be correct, and the benchmark seems to be constructed in a sound manner.

**Documentation:**

Yes

**Limitations:**

The authors address the limitations of their work well. However, I would like to know if they foresee any societal implications from their datasets.

**Opportunities For Improvement:**

- Even though, it is mentioned by the authors in the limitations. How does the sensitivity of the metrics vary with resolution of videos?

**Relation To Prior Work:**

Yes

**Summary And Contributions:**

The paper introduces a novel benchmark for time-lapse videos, ChronoMagic-Bench. The benchmark has 1649 prompts and realworld videos across 4 phenomenon. Further, two new metrics MTScore and CHScore are proposed to evaluate metamorphic attributes and temporal coherence in timelapse videos. Finally, the authors also release ChronoMagic-Pro - a large-scale dataset with 460k high-quality time-lapse videos and captions.

---

> ### Author Rebuttal · Authors · 2024-08-17
>
> Thanks for your constructive suggestions. Your endorsement of our benchmark and dataset gives us significant encouragement. Here are our clarifications.
> >**Q1**: Even though, it is mentioned by the authors in the limitations. How does the sensitivity of the metrics vary with resolution of videos?
>
> **A1**: Since most models do not support dynamic resolution and variable duration, it is not feasible to standardize these parameters. Consequently, we follow the official popular settings[1][2][3][4] to maintain the fairness. Moreover, both MTScore and CHScore mitigate the influence of resolution by employing a resizing strategy that adjusts the shorter edge and utilizes center cropping. MTScore further employs a fixed frame extraction method to ensure a consistent frame count, while the different terms of CHScore are insensitive to *num_frames*, thereby mitigating discrepancies due to varying frame numbers.
>
>
> >**Q2**: The authors address the limitations of their work well. However, I would like to know if they foresee any societal implications from their datasets.
>
> **A2**: Thank you for your thorough comments. We will add the following content to the latest version: "*After appropriate fine-tuning using the ChronoMagic-Pro/ProH dataset, text-to-video generation models are capable of producing not only metamorphic videos with extended time spans and high levels of realism but also high-quality general videos. These generative models could be misused to create deceptive videos, potentially misleading the public or disseminating misinformation, particularly on fast-paced and widely influential platforms such as social media.*"
>
> [1]Liu Y, Li L, Ren S, et al. Fetv: A benchmark for fine-grained evaluation of open-domain text-to-video generation[J]. Advances in Neural Information Processing Systems, 2024, 36.
>
> [2]Huang Z, He Y, Yu J, et al. Vbench: Comprehensive benchmark suite for video generative models[C]//Proceedings of the IEEE/CVF Conference on Computer Vision and Pattern Recognition. 2024: 21807-21818.
>
> [3]Wu J Z, Fang G, Wu H, et al. Towards a better metric for text-to-video generation[J]. arXiv preprint arXiv:2401.07781, 2024.
>
> [4]Sun K, Huang K, Liu X, et al. T2V-CompBench: A Comprehensive Benchmark for Compositional Text-to-video Generation[J]. arXiv preprint arXiv:2407.14505, 2024.

---

> > ### Comment · Reviewer_4KT6 · 2024-08-29
> >
> > Thank you authors for their response. I’ll retain my score.

---

### Official Review · Reviewer_JSoz · 2024-07-27
**Review for Submission 127**

**Rating:** 8
**Confidence:** 5
**Correctness:** N/A

**Review:**

**[Strength]**

S1. This study provides the evaluation prompts and training samples for time-lapse video clips having significant temporal changes and coherence.

S2. The idea of adopting visual tracking models to evaluate temporal coherence is interesting, and the results seem to be well-correlated with human evaluation.

S3. This study tried to conduct extensive experiments to benchmark existing video models.


**[Weakness]**

W1. The automatic metrics require more discussion and analysis to prove their efficacy.

W2. The details to ensure the systematic collection of data are missing.

W3. Experiments and their analysis should be improved.
W4. The checklist seems to include wrong information.

**Strengths:**

See the comments above.

**Additional Feedback:**

It’s not a major factor for the decision, but I leave some questions here.

Q1) In Line 114, the authors describe that 20 videos per subcategory are crawled for 75 categories, but I wonder why 1649 prompts are used for benchmark, not 1500 prompts? Does each category have a different number of prompts?

**Clarity:**

The explanation of automatic metrics should be improved, since the lack of details makes it hard for the readers to follow their details. The definition of mathematical terms is unclear. I recommend providing an intuitive example to explain the computation of automatic metrics proposed, defining each term in appropriate locations. For a nit example, note that $p_\text{vis}[i,j]$ is defined in Line 164, after introducing all equations in this page.

**Documentation:**

I could not find the example URLs of videos, although this study proposes datasets.

**Ethics:**

I could not find any description about data filtering for safety, bias, or privacy issues.

**Limitations:**

Although the authors address some limitations on their experiments, there is no discussion about the limitations of the data collection process and the collected data.

**Opportunities For Improvement:**

O1. The authors should clarify the efficacy of ChronoMagic-Pro. Specifically, OpenSoraPlan v1.1 [27] + ChronoMagic-Pro shows severe performance degradation in CHScore. In addition, its performance after fine-tuning is the worst performance in Table 3. Especially, note that the authors have claimed that CHScore is the most well-aligned metric with human evaluation in Figure 6. If ChronoMagic-Pro is effective, I think CHScore should be improved after finetune a model on the dataset.

---

O2. The explanation and analysis for CHScore can be improved.
- Which tracking algorithm is used to define the initial tracking points?
- What if some tracking points disappear in a natural time-lapse video? If there are many temporal changes, it is natural for some points to disappear overtime (e.g. sky time-lapse). Is CHScore robust to temporally coherent disappearance of points? How about there exist multiple objects with the same appearance?
- CHScore is defined by the inverse of the summation over the five metrics. However, why is the scale and impact of each metric considered equally? For example, the range of $R_\text{cut}$ and $M_\text{missed}$ could be different, since the numbers of frames and tracking points are different.

---

O3. The efficacy of proposed metrics can be provided more thoroughly. For example, could the authors provide qualitative analysis, in addition to quantitative analysis, where the case study gives the correct interpretation of a model’s performance?

---

O4. Considering the diversity of videos is one of the important parts for training data, ensuring the diversity of both ChronoMagic-Bench and ChronoMagic-Pro can improve the presentation of data quality.

---

O5. The data-preprocessing part could be explained or polished more.
- Is there any special reason to include 13% of ChronoMagic-Pro clips having low aesthetic scores?
- Why does ChronoMagic-Pro have an extremely large average length (234s) compared with other datasets including ChronoMagic (11.4s) in Table 2?
- Are the authors double-checked the overlap between ChronoMagic-Pro and ChronoMagic-Bench?
- How are the subcategories defined? Specifically, in “Biological” the level of subcategories are uneven. For example, although “butterflies” and “spider web” describe a specific entity, “animal” describes general classes. Are there any special reasons for it in a systematic manner?
- How can I assure the crawled video’s quality for ChronoMagic-Bench?

---

O6. The fair comparison and analysis of experiments could be clarified. Specifically, as described in the supplementary material, the generated videos of different models have different resolutions, frame rates, and durations. Could this setting deduce a fair conclusion? In addition, interpreting the open-sourced model’s performance, in terms of # of trainable parameters, and training data scale and flops, would be crucial to analyze each model’s performance.

---


O7. Human evaluation could be improved.
- Considering ChronoMagic-Bench includes 75 subcategories, 16 prompts, used for human evaluation, seems not to advocate the entire benchmark.
- there is no detailed explanation of voters.
- Considering the authors have used a simple five-scale evaluation, how are the answers post-processed to remove outliers?
- The authors claimed that MTScore is also correlated to human evaluation, but Figure 6 shows that MTScore cannot make a consistent ranking with human evaluation’s results.

---

O8. Analyzing the performance relationship between temporal changes/coherence and visual aesthetic would be worth exploration to improve the analyses.

---


O9. The authors should provide the correct information in the Checklist for the submission.
- 2-(a, b): there are no theoretical results/proofs in this paper, but “Yes” is checked.
- 3-(c): there is no error bar in this paper.
- 4-(a): there is no correct reference of the sourced video platform for each video or category.
- 4-(e): there is no mention of safety such as NSFW filtering.
- 5-(c): there is no mention of “the estimated hourly wage paid to participants” for human evaluation.

---

O10. The details for the video license should be provided more.
- In Line 542, the authors describe "We take full responsibility for the licensing, distribution, and maintenance of our ChronoMagic-Bench and ChronoMagic-Pro.” Are the licenses for all videos in both of two owned by the authors?
- What is the correct source of video crawling for each video? Can each video have different licenses (if the video licenses are not owned by the authors)?

**Relation To Prior Work:**

Yes.

**Summary And Contributions:**

This study aims to benchmark the existing T2V models’ capability to generate time-lapse videos having significant metamorphic amplitude and temporal coherence. Specifically, ChronoMagic-Bench consists of 1649 prompts in 75 subcategories for time-lapse videos. This study also proposes automatic metrics, Metamorphic Score (MTScore) and Temporal Coherence Score (CHScore), to evaluate metamorphic amplitude and temporal coherence, respectively. Finally, the ChronoMagic-Pro dataset is proposed to provide 460K video clips of high-quality time-lapse samples.

---

> ### Author Rebuttal · Authors · 2024-08-17
>
> Thank you for the time, thorough comments, and nice suggestions. We are pleased to clarify your questions step-by-step.
>
> >**Q1**: The authors should clarify the efficacy of ChronoMagic-Pro. Specifically, OpenSoraPlan v1.1 [27] + ChronoMagic-Pro shows severe performance degradation in CHScore. In addition, its performance after fine-tuning is the worst performance in Table 3. Especially, note that the authors have claimed that CHScore is the most well-aligned metric with human evaluation in Figure 6. If ChronoMagic-Pro is effective, I think CHScore should be improved after finetune a model on the dataset.
>
> **A1**: Thanks for your suggestions. First, due to limitations in computational resources, we selected only 10,000 video-text pairs from ChronoMagic-Pro for training, but the training did not fully converge before the deadline. Second, we initially employed a simple fine-tuning method with the primary goal of assessing the dataset's impact on enhancing metamorphic amplitude. However, this basic fine-tuning strategy may lead to flickering in the generated videos, as evidenced by a reduction in CHScore—a phenomenon confirmed by the *[MagicTime](https://github.com/PKU-YuanGroup/MagicTime)*. To address this issue, we adopted MagicTime's training strategy to fine-tune the OpenSoraPlan v1.1 model. The results (did not fully converge), as shown in the table below, demonstrate improvements across all metrics.
>
> | **Method**                                      | **Venue**  | **UMT-FVD ↓** | **UMTScore ↑** | **MTScore ↑** | **CHScore ↑** | **GPT4o-MTScore ↑** |
> |-------------------------------------------------|------------|---------------|----------------|---------------|---------------|----------------------|
> | OpenSoraPlan v1.1                         | Github'24  | 188.53        | 2.421          | 0.327         | 23.15     | 2.19                 |
> | OpenSoraPlan v1.1 + ChronoMagic-Pro 10K + simple finetune       | ours        | 185.72    | 2.753      | 0.341     | 15.62         | 3.03             |
> | OpenSoraPlan v1.1 + ChronoMagic-Pro 10K + magictime finetune       | ours        | **180.11**    | **2.864**  | **0.346**  |  **25.72**  | **3.05**             |
>
> >**Q2**:  Which tracking algorithm is used to define the initial tracking points?
>
> **A2**: We use Meta AI Research's *[CoTrack](https://github.com/facebookresearch/co-tracker)* to define the initial tracking points, as indicated in line 153 of the main text.
>
> >**Q3**: What if some tracking points disappear in a natural time-lapse video? If there are many temporal changes, it is natural for some points to disappear overtime (e.g. sky time-lapse). Is CHScore robust to temporally coherent disappearance of points? How about there exist multiple objects with the same appearance?
>
> **A3**: For the first question, we have updated the computation to make the CHScore robust to temporally coherent disappearance of points. Specifically, we first calculate the direction of camera/object movement based on the tracking points across all frames. Then, if the tracking point *j* of frame *i* disappears in the far direction, it is not included in the calculation of *m[i]*. The results are shown in the table below. As can be seen, the results are basically consistent with those before the improvement, because the existing text-to-video generation models produce relatively static videos and are unable to generate content with a large temporal span.
>
> |  | ModelScopeT2V | ZeroScope | T2V-zero | LaVie | AnimateDiff | VideoCrafter2 | MagicTime | Latte | OpenSora 1.1 | OpenSoraPlan v1.1 |
> |---------|---------------|-----------|----------|-------|-------------|---------------|-----------|-------|--------------|-------------------|
> | **CHScore**                    | 11.03 | 25.13 | 1.68 | 8.60 | 11.36 | 8.27 | 10.66 | 13.81 | 10.03 | 10.35 |
> | **CHScore (+direction)** | 11.42 | 25.28 | 1.71 | 8.62 | 11.43 | 8.53 | 11.08 | 14.13 | 10.34 | 10.60 |
>
> For the second question, CoTrack tracks objects based on pixel position features, allowing it to function correctly even with multiple objects of the same appearance, as different objects have distinct positional features. So it does not affect the calculation of CHScore.

---

> > ### Author Rebuttal · Authors · 2024-08-17
> >
> > >**Q4**: CHScore is defined by the inverse of the summation over the five metrics. However, why is the scale and impact of each metric considered equally? For example, the range of *R_cut* and *M_missed* could be different, since the numbers of frames and tracking points are different.
> >
> > **A4**: "Tracking points" are fixed, and "the number of frames" has a relatively small impact on the results because the term is divided by *num_frames*. Additionally, we have enhanced the accuracy of CHScore by standardizing each term to a range of 0 to 1 and adjusting the weighting of each term. The updated results are available on the *[Leaderboard](https://huggingface.co/spaces/BestWishYsh/ChronoMagic-Bench)*. Specifically, *R_missed* is a global metric representing the model's overall performance across the entire video and holds the highest significance, with a weight of 0.35. *V_missed* measures the stability of missed points between frames, a critical aspect of video analysis, and is therefore assigned a weight of 0.25. *R_cut* indicates abnormal situations and carries a weight of 0.15. *C_missed*, similar in function to the *R_cut*, serves as a secondary indicator, also weighted at 0.15. Lastly, *M_missed* represents individual extreme cases and is assigned a lower weight of 0.10. The updated *[human evaluation](https://pku-yuangroup.github.io/ChronoMagic-Bench/static/images/human_evaluation.jpg)* demonstrates that the improved CHScore aligns more closely with human perception.
> >
> > |  | ModelScopeT2V | ZeroScope | T2V-zero | LaVie | AnimateDiff | VideoCrafter2 | MagicTime | Latte | OpenSora 1.1 | OpenSoraPlan v1.1 |
> > |---------|---------------|-----------|----------|-------|-------------|---------------|-----------|-------|--------------|-------------------|
> > | **CHScore**                    | 11.03 | 25.13 | 1.68 | 8.60 | 11.36 | 8.27 | 10.66 | 13.81 | 10.03 | 10.35 |
> > | **CHScore (+direction)** | 11.42 | 25.28 | 1.71 | 8.62 | 11.43 | 8.53 | 11.08 | 14.13 | 10.34 | 10.60 |
> > | **CHScore (+direction & weight)** | 29.23 | 46.13 | 8.62 | 28.01 | 30.35 | 27.80 | 29.03 | 34.73 | 25.34 | 23.15 |
> >
> > >**Q5**: The efficacy of proposed metrics can be provided more thoroughly. For example, could the authors provide qualitative analysis, in addition to quantitative analysis, where the case study gives the correct interpretation of a model’s performance?
> >
> > **A5**: Thanks for your nice suggestions. We have provided qualitative results of MTScore and CHScore in Appendix Section B.3. We also provide additional quantitative and qualitative results of videos generated by different models, which can better validate the efficacy of the proposed metrics. Please refer specifically to the following sections on the [_Page_](https://pku-yuangroup.github.io/ChronoMagic-Bench): "Evaluation Examples of Different Models," "Metamorphic Amplitude References," and "Temporal Coherence References."
> >
> > >**Q6**: Considering the diversity of videos is one of the important parts for training data, ensuring the diversity of both ChronoMagic-Bench and ChronoMagic-Pro can improve the presentation of data quality.
> >
> > **A6**: ChronoMagic-Bench and ChronoMagic-Pro are constructed using a search database that includes 75 categories. These categories are selected through frequency-based filtering of an extensive list of search terms, ensuring each category is represented by a sufficient number of videos. Moreover, the video authors are from diverse global locations, thereby ensuring diversity.
> >
> > >**Q7**: Is there any special reason to include 13% of ChronoMagic-Pro clips having low aesthetic scores?
> >
> > **A7**: Actually, the *[aesthetic detector](https://github.com/LAION-AI/aesthetic-predictor)* inherently possesses biases, favoring artistic images, such as oil paintings and other art forms, over realistic styles. Therefore, low aesthetic scores do not necessarily signify poor data. Retaining a small portion of such data can contribute to enhancing the diversity of the videos.
> >
> > >**Q8**: Why does ChronoMagic-Pro have an extremely large average length (234s) compared with other datasets including ChronoMagic (11.4s) in Table 2?
> >
> > **A8**: There are two types of time-lapse videos: compressed and uncompressed. The former represents the entire process in a few seconds, while the latter can last for several minutes or even tens of minutes. ChronoMagic consists of compressed videos, whereas ChronoMagic-Pro includes both types to increase diversity. If the 60s+ videos, which account for 27% of ChronoMagic-Pro, are excluded, the average length would be only 12.36 seconds.
> >
> > >**Q9**: Are the authors double-checked the overlap between ChronoMagic-Pro and ChronoMagic-Bench?
> >
> > **A9**: Yes, we deduplicate using video IDs. Moreover, the labeling models used by the two are different, so even if the videos are identical, the ChronoMagic-Bench results remain unaffected, as the evaluation is based on prompts, with the videos serving as references.
> >
> > >**Q10**: How are the subcategories defined? Specifically, in “Biological” the level of subcategories are uneven. For example, although “butterflies” and “spider web” describe a specific entity, “animal” describes general classes. Are there any special reasons for it in a systematic manner?
> >
> > **A10**: We manually define the subcategories with the guidence by GPT-4o. This uneven approach can keep the classification system flexible: broad categories make it easier for users to conduct general searches, while specific categories help users find more precise information in a particular area. Actually, search engines retrieve videos based on titles and tags. Occasionally, a search for 'animal' may not produce results for 'butterflies,' and vice versa. Therefore, subcategories encompass both specific terms, such as 'butterflies,' and general ones, like 'animal,' to ensure comprehensive search results and preserve the dataset's diversity.

---

> > ### Author Rebuttal · Authors · 2024-08-17
> >
> > >**Q11**: How can I assure the crawled video’s quality for ChronoMagic-Bench?
> >
> > **A11**: All videos in ChronoMagic-Bench are manually crawled. We exclusively select videos with a distinctly observable timelapse process. Additionally, we screen videos using the metadata provided by the video platform. Videos with a low number of likes, downloads, or views, as well as those with short titles, inadequate tags, or tags containing terms such as 'YouTube,' 'video,' and 'shorts,' are excluded from our collection.
> >
> > >**Q12**: The fair comparison and analysis of experiments could be clarified. Specifically, as described in the supplementary material, the generated videos of different models have different resolutions, frame rates, and durations. Could this setting deduce a fair conclusion? In addition, interpreting the open-sourced model’s performance, in terms of # of trainable parameters, and training data scale and flops, would be crucial to analyze each model’s performance.
> >
> > **A12**: For the first issue, since most T2V models do not support dynamic resolution or variable duration, it is not feasible to standardize these parameters. Therefore, we follow the official popular settings[1][2][3][4][5][6] to maintain a degree of fairness. Additionally, MTScore and CHScore implement short-side resizing and center-cropping techniques to mitigate the effects of resolution variability. MTScore further employs a fixed frame extraction method to ensure a consistent frame count, while the different terms of CHScore are insensitive to *num_frames*, thereby mitigating discrepancies due to varying frame numbers.
> >
> > For the second issue, we are also unable to obtain details of  #. For instance, ModelScoreT2V and ZeroScope provide only inference APIs without corresponding source code, making it impossible to verify trainable parameters and flops. Similarly, OpenSoraPlan, OpenSora, and LaVie utilize custom-built datasets but do not disclose specific training data scale, such as the number of videos or total duration.
> >
> > > **Q13**: Considering ChronoMagic-Bench includes 75 subcategories, 16 prompts, used for human evaluation, it seems not to advocate the entire benchmark.
> >
> > **A13**: First, the 16 prompts were carefully selected to reflect the overall situation to a certain extent, with the number being based on references [1][2][3][4][5][6]. Second, despite the limited number of prompts, completing each questionnaire demands between 10 to 20 minutes, indicating a significant time investment. Furthermore, to better advocate the entire benchmark, we have recruited 41 additional voters, conducting a further human evaluation on OpenSora 1.2 and Reference Videos. The results, detailed on the *[Page](https://pku-yuangroup.github.io/ChronoMagic-Bench/)* titled '*[Validation of the Automatic Metrics](https://pku-yuangroup.github.io/ChronoMagic-Bench/static/images/human_evaluation.jpg)*,' demonstrate that the proposed metrics align with human perception.
> >
> > >**Q14**: There is no detailed explanation of voters.
> >
> > **A14**: The voter population predominantly comprises undergraduate, master's, and phd students from universities, along with a segment of the general public who are not associated with this field. They come from various regions around the world, including China, USA, Singapore, etc., which ensures that the participants have universality. This composition guarantees the precision and diversity of human evaluations.
> >
> > >**Q15**: Considering the authors have used a simple five-scale evaluation, how are the answers post-processed to remove outliers?
> >
> > **A15**: We remove outliers of the answer as follows:
> > - Restricted each IP address to prevent duplicates and required users to log in to their accounts before voting, ensuring that each person can only submit once.
> > - Determined the validity of data based on the time taken to complete the questionnaire. Given that completing a questionnaire typically takes 10 to 20 minutes, we excluded samples where the response time was less than 10 minutes.
> > - Randomized the order in which different videos were presented to avoid cognitive biases among voters.
> > - Required a sliding verification at submission to ensure that all questionnaires were completed manually and not by bots.
> > - Discarded any questionnaire where 50% of the ratings were extreme values, i.e., **the sum of 5-point and 1-point options exceeded 50%.**
> >
> > >**Q16**: The authors claimed that MTScore is also correlated to human evaluation, but Figure 6 shows that MTScore cannot make a consistent ranking with human evaluation’s results.
> >
> > **A16**: We apologize for any confusion caused. Initially, the outliers were not processed, resulting in inaccuracies. The updated *[results](https://pku-yuangroup.github.io/ChronoMagic-Bench/static/images/human_evaluation.jpg)*, available on the *[Page](https://pku-yuangroup.github.io/ChronoMagic-Bench/)*, demonstrate consistency between the metrics and human perception.
> >
> >
> > [1]Liu Y, Li L, Ren S, et al. Fetv: A benchmark for fine-grained evaluation of open-domain text-to-video generation[J]. Advances in Neural Information Processing Systems, 2024, 36.
> >
> > [2]Huang Z, He Y, Yu J, et al. Vbench: Comprehensive benchmark suite for video generative models[C]//Proceedings of the IEEE/CVF Conference on Computer Vision and Pattern Recognition. 2024: 21807-21818.
> >
> > [3]Wu J Z, Fang G, Wu H, et al. Towards a better metric for text-to-video generation[J]. arXiv preprint arXiv:2401.07781, 2024.
> >
> > [4]Sun K, Huang K, Liu X, et al. T2V-CompBench: A Comprehensive Benchmark for Compositional Text-to-video Generation[J]. arXiv preprint arXiv:2407.14505, 2024.
> >
> > [5]Singer U, Polyak A, Hayes T, et al. Make-A-Video: Text-to-Video Generation without Text-Video Data[C]//The Eleventh International Conference on Learning Representations.
> >
> > [6]Wu J Z, Fang G, Wu H, et al. Towards a better metric for text-to-video generation[J]. arXiv preprint arXiv:2401.07781, 2024.

---

> ### Author Rebuttal · Authors · 2024-08-17
>
> >**Q17**: Analyzing the performance relationship between temporal changes/coherence and visual aesthetic would be worth exploration to improve the analyses.
>
> **A17**: Thanks for your constructive suggestions. In fact, one major function of the CHScore is to measure the visual aesthetic of metamorphic videos. A high MTScore value only indicates whether the temporal changes are sufficiently large, whereas the introduction of CHScore helps to assess whether these temporal changes are smooth enough, that is, whether the visual aesthetic is satisfactory. The combined use of both metrics allows for a comprehensive evaluation of the quality of time-lapse videos.
>
> >**Q18**: The authors should provide the correct information in the Checklist for the submission.
> >- 2-(a, b): there are no theoretical results/proofs in this paper, but “Yes” is checked.
> >- 3-\(c): there is no error bar in this paper.
> >- 4-(a): there is no correct reference of the sourced video platform for each video or category.
> >- 4-(e): there is no mention of safety such as NSFW filtering.
> >- 5-\(c): there is no mention of “the estimated hourly wage paid to participants” for human evaluation.
>
> **A18**: We are sorry for the error in filling out the CheckList, and will make the following clarification in the latest version:
> - 2-(a, b): We will change the option to "No".
> - 3-\(c): We will change the option to "No". Actually, we mention that the results of the T2V generation model vary depending on the random seed. Therefore, we ran the model three times for each prompt and averaged the scores from the three runs to obtain the final scores.
> - 4-(a): ChronoMagic-Bench videos mainly come from Pexels, MixKit, PixaBay and YouTube. ChronoMagic-Pro videos are all from YouTube.
> - 4-(e): We will add the answer of **Q22** in the final version
> - 5-\(c): We regard an individual’s hourly wage or compensation as personal information, which, due to privacy considerations, cannot be disclosed. Nonetheless, we can confirm that all participants received appropriate compensation in compliance with the legal requirements of their respective countries or regions.
>
> >**Q19**: The details for the video license should be provided more.
> >- In Line 542, the authors describe "We take full responsibility for the licensing, distribution, and maintenance of our ChronoMagic-Bench and ChronoMagic-Pro.” Are the licenses for all videos in both of two owned by the authors?
> >- What is the correct source of video crawling for each video? Can each video have different licenses (if the video licenses are not owned by the authors)?
>
> **A19**: The videos utilized by ChronoMagic-Bench is sourced from free content available on four platforms: Pexels (CC0), MixKit (CC0), PixaBay (CC0), and YouTube (CC BY 4.0). Conversely, ChronoMagic-Pro exclusively employs videos from YouTube (CC BY 4.0). The licensing types of these videos are clearly indicated on their respective platforms. The CC0 license (Creative Commons Zero) designates content as public domain, allowing unrestricted use without the need for additional permissions or licenses. Videos from the YouTube platform adhere to the CC BY 4.0 license (Creative Commons Attribution 4.0); consequently, we have included video IDs and author information in the metadata to prevent any potential contractual disputes.
>
> >**Q20**: Although the authors address some limitations on their experiments, there is no discussion about the limitations of the data collection process and the collected data.
>
> **A20**: For the "data collection process," the majority of the data is sourced from the YouTube platform, where video quality varies significantly, with only a small portion coming from higher-quality platforms like Pexels, MixKit, and PixaBay, due to the limited availability of time-lapse videos on the latter. This necessitates spending a significant amount of time filtering the videos, for example, based on aesthetic criteria, views, likes, and other factors. As for the "collected data," since most of the videos are licensed under CC BY 4.0 rather than CC0, the data is restricted to academic research use and cannot be used for commercial purposes.

---

> > ### Author Rebuttal · Authors · 2024-08-17
> >
> > >**Q21**: I could not find the example URLs of videos, although this study proposes datasets.
> >
> > **A21**: Sorry for the inconvenience, we have uploaded all the relevant content:
> >
> > *[**Page**](https://pku-yuangroup.github.io/ChronoMagic-Bench):https://pku-yuangroup.github.io/ChronoMagic-Bench*
> >
> >  *[**Code**](https://github.com/PKU-YuanGroup/ChronoMagic-Bench):https://github.com/PKU-YuanGroup/ChronoMagic-Bench*
> >
> > *[**Dataset & Benchmark**](https://huggingface.co/collections/BestWishYsh/chronomagic-bench-667bea7abfe251ebedd5b8dd):https://huggingface.co/collections/BestWishYsh/chronomagic-bench-667bea7abfe251ebedd5b8dd*.
> >
> > >**Q22**: I could not find any description about data filtering for safety, bias, or privacy issues.
> >
> > **A22**: The video content consists entirely of time-lapse footage, and we detect and discard NSFW content based on the video captions. For videos involving identifiable individuals, we accelerate the blurring process to ensure the security of personally identifiable information. The collected videos are organized into four major categories (comprising 75 subcategories), with creators hailing from various countries and regions worldwide. This diversity ensures that ChronoMagic-Bench and ChronoMagic-Pro possess ample representativeness. The Open-Sora-Plan model, fine-tuned using our dataset, exhibited no significant content bias.
> >
> > >**Q23**: It’s not a major factor for the decision, but I leave some questions here. In Line 114, the authors describe that 20 videos per subcategory are crawled for 75 categories, but I wonder why 1649 prompts are used for benchmark, not 1500 prompts? Does each category have a different number of prompts?
> >
> > **A23**: Apologies for the confusion. Each subcategory contains at least 20 videos, with the number of videos corresponding to general terms typically being greater than those for specific terms, as the former encompasses a wider range of variation.

---

> ### Author Rebuttal · Authors · 2024-08-24
>
> Dear Reviewer JSoz,
>
> We are deeply grateful for your attention and care to our work. Understanding the importance of thorough feedback, we're here to address any queries or points of ambiguity regarding our response. Please feel free to reach out with any further questions.
>
> Best,
> Authors

---

> > ### Comment · Reviewer_JSoz · 2024-08-28
> >
> > I sincerely appreciate your addressing my concerns. Many concerns are addressed by authors responses during the rebuttal period, and they should be added into the revised version.
> >
> > > However, this basic fine-tuning strategy may lead to flickering in the generated videos, as evidenced by a reduction in CHScore—a phenomenon confirmed by the MagicTime. To address this issue, we adopted MagicTime's training strategy to fine-tune the OpenSoraPlan v1.1 model. The results (did not fully converge), as shown in the table below, demonstrate improvements across all metrics.
> >
> > The major blocker for my confident decision is this result. Could the authors specify what "MagicTime's training strategy" is? Is the flickering artifacts common for all dataset, or tailored issue for the proposed dataset? Could the authors further verify that the improvements in CHScore by simple fine-tuning with small subset or other models?

---

> > > ### Comment · Reviewer_JSoz · 2024-08-28
> > >
> > > In addition, could the author provide some explanation or revision sketch to address my concerns below ?
> > > > W1. The automatic metrics require more discussion and analysis to prove their efficacy.

---

> ### Author Rebuttal · Authors · 2024-08-28
>
> Thanks for your reply. We have added all the discussed content to the revised version.
>
> >**Add Q1**: The major blocker for my confident decision is this result.
> >
> >**(a)** Could the authors specify what "MagicTime's training strategy" is?
> >
> >**(b)** Is the flickering artifacts common for all dataset, or tailored issue for the proposed dataset?
> >
> >**(c)** Could the authors further verify that the improvements in CHScore by simple fine-tuning with small subset or other models?
>
> **Add A1**: We sincerely appreciate your thoughtful feedback! Below are our detailed responses:
>
> **(a)** The Magic Training Strategy [1] introduces an innovative approach tailored for fine-tuning pretrained text-to-video models using time-lapse videos. This strategy effectively incorporates time-lapse video priors while retaining general video priors, ultimately facilitating the generation of videos with high temporal coherence and minimal flickering.
>
> This method employs a two-stage training stage that decouples the spatial and temporal layers, thereby reducing the complexity associated with learning from time-lapse videos. In **stage one**, the temporal layer is removed from the pretrained model, and the **MagicAdapter-S** is introduced, which is trained on keyframe-text pairs to encode static state information. In **stage two**, the temporal layer is reintroduced, and the **MagicAdapter-T** is integrated, which is trained on entire video-text pairs to capture dynamic state changes. Moreover, the strategy employs a dynamic frame extraction technique, opting for either uniform or random sampling based on the video’s motion magnitude, thus significantly bolstering the model's resilience to both general and time-lapse video types.
>
> By employing this training strategy, we can generate videos with enhanced temporal coherence and reduced flickering.
>
> [1] Yuan, Shenghai, et al. "MagicTime: Time-lapse Video Generation Models as Metamorphic Simulators." *arXiv preprint arXiv:2404.05014* (2024).
>
> **(b)** Flickering artifacts are indeed a common challenge across all T2V datasets and models. The underlying causes are multifaceted, including not only dataset characteristics but also training strategies and model architectures. Initially, our use of a static frame selection strategy resulted in training frames prone to flickering. Moreover, a basic fine-tuning approach made it challenging to internalize new concepts such as time-lapse motion patterns, leading to a conflation of general and time-lapse video processing, which further exacerbated flickering. The Magic Training Strategy, however, provides an effective solution to mitigate these issues.
>
> **(c)** We have conducted fine-tuning of the OpenSoraPlan v1.1 model using either a simple fine-tuning strategy or the Magic Training Strategy on the 10K ChronoMagic-Pro subset. The results demonstrated that the magic training strategy improved the CHScore from 23.15 to 25.72, underscoring its effectiveness when applied to the ChronoMagic-Pro dataset.
>
> In addition, as shown in Figure 12 of [latest version](https://drive.google.com/file/d/1KomuKO9pPBoyGzThV8uIhe0GGYx9lgbV/view?usp=sharing), we show the qualitative experiment of OpenSoraPlan v1.1 among origin, simple fine-tuning and magic training, which further verifies the improvement of the temporal coherence of the videos after magic training.
>
> Given that the discussion period concludes on August 31, 2024, our current computational and time resources are likely insufficient to support additional training of other computationally intensive text-to-video models, which typically require 4-5 days for fine-tuning on an 8xA100/H100 setup.
>
> Nevertheless, we promise to include additional experiments in the final version, examining the performance of various T2V models (e.g., Animatediff, OpenSoraPlan) using either simple fine-tuning or the magic training strategy across different ChronoMagic-Pro subsets, to validate improvements in CHScore.

---

> ### Author Rebuttal · Authors · 2024-08-28
>
> >**Add Q2**: In addition, could the author provide some explanation or revision sketch to address my concerns below?
> >
> >**W1**. The automatic metrics require more discussion and analysis to prove their efficacy.
>
> **Add A2**:  Following your suggestion, as shown in the [latest version](https://drive.google.com/file/d/1KomuKO9pPBoyGzThV8uIhe0GGYx9lgbV/view?usp=sharing) and [page](https://pku-yuangroup.github.io/ChronoMagic-Bench), we have conducted a series of additional quantitative and qualitative analyses to validate the proposed metrics which may address your concerns.
>
> Specifically, all the explanations and revision sketches are available below:
>
> ***Page***: [https://pku-yuangroup.github.io/ChronoMagic-Bench](https://pku-yuangroup.github.io/ChronoMagic-Bench)
>
> (*Metamorphic Amplitude References*)
>
> (*Temporal Coherence References*)
>
> (*Validation of the Automatic Metrics*)
>
> (*Evaluation examples of Different Models* with Different MTScore and CHScore)
>
> ***Latest Paper***: https://drive.google.com/file/d/1KomuKO9pPBoyGzThV8uIhe0GGYx9lgbV/view?usp=sharing
>
> (*Revised Main body for describing CHScore* in Lines 151-170)
>
> (*Appendix C.3 Visual Reference of the Different Scores of MTScore and CHScore* and Figure 7 in Lines 691-694)
>
> (*Appendix C.2 Further Description of Temporal Coherence Score* in Lines 660-690)
>
> (*Appendix E.2 Details of Evaluation Models* for MTScore and CHScore in Lines 762-769)
>
> (*Appendix Figure 15: Alignment between automatic metrics and human perception in terms of disaggregated data*)
>
> (*Appendix E.6 Additional Details of Human Evaluation* in Lines 919-947)
>
> Please let us know if there are further questions.

---

> > ### Comment · Reviewer_JSoz · 2024-08-29
> >
> > I appreciate the additional responses. Based on the responses during rebuttal, I have changed my initial rating. Hopefully, I strongly recommend to add the discussion details in the revised version, since the discussions can improve the quality and impacts of this study.

---

> > > ### Author Rebuttal · Authors · 2024-08-29
> > >
> > > Thank you for the quick reply and constructive suggestions! We promise to add the discussion details in the revised version (camera-ready version).

---

### Author Response · Authors · 2024-08-27
**General Response to ACs, Reviewers, and Ethics Reviewers**

We sincerely thank the reviewers for their detailed and valuable feedback. All reviewers **(JSoz, 4KT6, qu3H, ntFp)** acknowledged the significant contributions this study makes to the evaluation of T2V models. They highlighted the innovative introduction of the ChronoMagic-Bench and the novel metrics (e.g., CHScore and MTScore), which align well with human perception and address a critical gap in existing T2V benchmarks **(4KT6, qu3H, ntFp)**. The reviewers appreciated extensive experiments conducted **(JSoz, qu3H, ntFp)** as well as the clarity and organization of the paper **(4KT6, ntFp)**.

Based on these comments, we have provided the following noteworthy responses:

- **[Ethics Reviewer VcuK, sLoc; Reviewers JSoz, 4KT6, qu3H, ntFp]** We have revisited the ethics checklist and supplemented the main text with sections on '*Potential Harms Caused by the Research Process*,' '*Societal Impact and Potential Harmful Consequences*,' and  '*Impact Mitigation Measures*.' Additionally, we have included a detailed explanation of the voters' background.

- **[Reviewers JSoz, 4KT6, qu3H, ntFp]** We have made all relevant content publicly accessible to ensure rigor and reproducibility: *[Page](https://pku-yuangroup.github.io/ChronoMagic-Bench)*, *[Code](https://github.com/PKU-YuanGroup/ChronoMagic-Bench)*, and *[Dataset & Benchmark](https://huggingface.co/collections/BestWishYsh/chronomagic-bench-667bea7abfe251ebedd5b8dd)*.

- **[Reviewers JSoz, ntFp]** We have clarified that the low CHScore for **OpenSoraPlan v1.1 + ChronoMagic-Pro 10K** was due to an overly simplistic and insufficiently rigorous training strategy. A more advanced training strategy has been adopted, confirming the effectiveness of ChronoMagic-Pro.

- **[Reviewers JSoz, ntFp]** We have improved the CHScore by assigning more appropriate weighting coefficients to each term and utilizing the direction of camera/object movement to mitigate the impact of temporal coherence loss, resulting in a metric more aligned with human perception.

- **[Reviewers JSoz, qu3H, ntFp]** We have conducted additional quantitative and qualitative experiments to validate the proposed metrics and dataset. These experiments included evaluating a broader range of open-source and closed-source models, providing more detailed per-video metric scores, conducting human evaluations on individual videos, and fine-tuning ChronoMagic-Pro with various training strategies, as well as providing more model-generated cases.

- **[Reviewers JSoz, qu3H]** We have clarified the diversity of the proposed dataset and benchmark consisting of 75 diverse categories, with contributions from global video creators.

- **[Reviewers JSoz, 4KT6]** We have explained that the comparison and analysis of experiments are fair, as most models do not support dynamic resolution and variable duration, and both CHScore and MTScore have implemented measures to address this issue.

- **[Reviewers JSoz, qu3H, ntFp]** We have conducted additional human evaluations on real videos and more T2V models, and removed outliers from the questionnaires, further enhancing the reliability of the evaluation.

- **[Reviewer qu3H, ntFp, ntFp]** We have polished this paper and fixed the grammatical errors, quotations, and expressions in the revised version.

- **[Reviewer JSoz]** We have clarified misunderstandings regarding ChronoMagic-Pro and ChronoMagic-Bench, such as why 13% of the videos have low aesthetic quality, why the average video length is 234 seconds, how subcategories are defined, whether videos are filtered for quality, and whether there is overlap in the dataset and benchmark.

- **[Reviewer qu3H]** We have explained the rationale for using GPT-4o as the evaluation criterion in the article.

- **[Reviewer ntFp]** We have clarified that the paper is broadly relevant to the community, as it can address the tendency of many video models to generate nearly static videos for various topics by fine-tuning on the proposed dataset.

- **[Reviewer ntFp]** We have provided an option for video authors to remove their content from the dataset via [GitHub](https://github.com/PKU-YuanGroup/ChronoMagic-Bench).

- **[Reviewer ntFp]** We have clarified that the template "An x video, not a y video" works better than "An x video" for InternVideo2.

- **[Reviewer ntFp]** We have discussed potential future directions, including using more advanced MLLMs to help evaluate video generation models.

We sincerely hope this work offers valuable insights into the field of video generation. Thanks again to all reviewers for their valuable time to help improve our work.

---

### Decision · Program_Chairs · 2024-09-26

**Decision:**

Accept (Spotlight)

**Comment:**

The paper introduces a benchmark for the evaluation of time-lapse videos, filling a gap in the evaluation of T2V models.

Strenghts:
* The proposed metrics, MTScore and CHScore, are designed to measure metamorphic amplitude and temporal coherence, respectively, and are well correlated with human perception according to the experiments.
* The benchmark includes a thorough evaluation of existing open-source models, offering insights into their strengths and weaknesses.
* The ChronoMagic-Pro dataset provides a valuable new T2V dataset focusing on time-lapse videos.

Weaknesses:
* The initial evaluation was restricted to open-source models. To address this, the authors expanded their evaluation to include closed-source models like Gen-2, Pika, KeLing, and LUMA.
* The proposed metrics require more discussion and analysis to prove their efficacy. The authors addressed this by including more experiments.
* The ethics review revealed points that had not been addressed relating to usage rights, bias and safeguards -- which the authors have thoroughly addressed.

The paper was well received among all of the reviewers, and the authors have provided a very thorough rebuttal to address concerns. We recommend spotlight acceptance.